



# The dual-field-of-view polarization lidar technique: A new concept in monitoring aerosol effects in liquid-water clouds — Theoretical framework

Cristofer Jimenez[1], Albert Ansmann[1], Ronny Engelmann[1], David Donovan[2], Aleksey Malinka[3], Jörg Schmidt[4], Patric Seifert[1], and Ulla Wandinger[1]

[1]Leibniz Institute for Tropospheric Research, Leipzig, Germany
[2]Royal Netherlands Meteorological Institute (KNMI), De Bilt, The Netherlands
[3]National Academy of Sciences of Belarus, Minsk, Belarus
[4]Institute of Meteorology, University of Leipzig, Leipzig, Germany

**Correspondence:** Cristofer Jimenez (jimenez@tropos.de)

**Abstract.**

In a series of two articles, a novel, robust, and practicable lidar approach is presented that allows us to derive microphysical properties of liquid-water clouds (cloud extinction coefficient, droplet effective radius, liquid-water content, cloud droplet number concentration) at a height of 50-100 m above cloud base. The temporal resolution of the observations is on the order of
30-120 sec. Together with the aerosol information (aerosol extinction coefficients, cloud condensation nucleus concentration) below the cloud layer, obtained with the same lidar, in-depth aerosol-cloud interaction studies can be performed. The theoretical background and the methodology of the new cloud lidar technique is outlined in this article (part 1), measurement applications are presented in a companion publication (part 2). The novel cloud retrieval technique is based on lidar observations of the volume linear depolarization ratio at two different receiver field-of-views (FOVs). Extensive simulations of lidar returns in
the multiple scattering regime were conducted to investigate the capabilities of a dual-FOV polarization lidar to measure cloud properties and to quantify the information content in the measured depolarization features regarding the basic retrieval parameters (cloud extinction coefficient, droplet effective radius). Key simulation results and the developed overall data analysis scheme to obtain the aerosol and cloud products are presented.

# 1 Introduction

Aerosol-cloud-preciptation interaction is an important branch of atmospheric research and one of the main uncertainty sources in climate predictions (IPCC, 2014). Strong efforts are undertaken to investigate the role of aerosols in liquid-water, mixed-phase, and cirrus cloud formation processes, by means of ground-based, airborne, and spaceborne observations with an increasing contribution of active remote sensing (Grosvenor et al., 2018). Ground-based lidar is the most favorable technique





to continuously monitor aerosol layers and the evolution of clouds within these layers. Regarding liquid-water clouds, lidar permits us to measure aerosol properties directly below cloud base and liquid-droplet microphysical properties just above cloud base and thus to quantify the relationship between changing aerosol conditions and changing cloud properties very sensitively and with high temporal resolution. The impact of up and downward motions which strongly influence the levels of water vapor

supersaturation during droplet formation and thus control how many of the aerosol particles will be activated the become cloud droplets, can be investigated in these aerosol-cloud-interaction (ACI) studies by adding or integrating a vertically pointing Doppler lidar to the remote sensing facility (Schmidt et al., 2014, 2015).

The new dual-FOV polarization lidar technique, introduced in this article, is a follow-up development of the dual-FOV Raman lidar technique (Schmidt et al., 2013) which allows us to determine the effective radius of cloud droplets and the cloud

light-extinction coefficient, and to derive the liquid water content and cloud droplet number concentration within the lowest 100 m of a liquid-water cloud layer. Together with aerosol properties such as the particle extinction coefficient or the estimated cloud condensation nucleus (CCN) concentration in air parcels, that enter the cloud environment in updrafts from below, the influence of aerosol particles on the evolution of the cloud layer can be monitored in large detail.

Lidar observations of liquid-water cloud properties make use of the relationship between the strength of multiple scattering

caused by water droplets and the size and amount of these droplets. In the case of the dual-FOV Raman lidar technique, nitrogen Raman backscatter signals are measured at two different receiver FOVs to provide the necessary information about multiple scattering. The advantage of the Raman lidar is that the measured multiple scattering contribution (forward scattering of laser photons by cloud droplets) is unambiguously linked to the effective radius of the droplets. This method delivers the most robust and reliable observations of microphysical properties of liquid-water clouds. However, nitrogen Raman signals are

weak so that observations are restricted to nighttime hours and signal averaging times of 10-30 minutes are usually needed to reduce the impact of signal noise on the lidar products to a tolerable level. Thus, the investigation of the influence of aerosols on the evolution of the cloud system with high resolution of seconds to minutes at day and nighttime is not possible with the Raman lidar. Furthermore, because of these long signal integration times a bias in the retrieval products caused by averaging of backscatter signals during periods with a varying cloud base height resulting from up and downward motions must be kept in

consideration in the data interpretation (Schmidt et al., 2013, 2014). This problem is widely overcome in the case of the novel dual-FOV polarization lidar technique with respective short signal integration times.

The requirement for observations at day and nighttime and temporal resolutions of the order of 30-120 s to resolve different phases of cloud evolution and to study, e.g., the impact of individual updraft events of given duration and strength on cloud droplet nucleation for given aerosol conditions was however the main motivation for the development of this alternative lidar

measurement concept (Jimenez et al., 2017, 2018). A polarization lidar transmits linearly polarized laser pulses and detects the cross- and co-polarized signal components. "Co" and "cross" denote the planes of polarization parallel and orthogonal to the plane of linear polarization of the transmitted laser pulses, respectively. The volume linear depolarization ratio is defined as the ratio of the cross- to the co-polarized signal and yields the information on the ratio of the cross-to-co-polarized backscatter coefficient. The depolarization ratio is sensitively influenced by multiple scattering in water clouds and varies, e.g., with

receiver FOV, cloud height, and number concentration and size of the droplets as will be explained in this article. Comparably





strong cloud elastic-backscatter signals are the basis for this methods so that no restrictions to nighttime hours are given and a high temporal resolution can be achieved. The light-depolarizing effect is different for different FOVs and this difference sensitively depends on the effective radius of the droplets. The strength of the change in light depolarization with height inside the cloud layer provides a direct measurement of the cloud light-extinction coefficient. All this is outlined in Sect. 3.

The article is organized as follows: In Sect. 2, a brief review of lidar methods for liquid-water cloud observations is given. Sect. 3 provides the theoretical background regarding the relationship between multiple scattering effects as a function of the microphysical properties of the liquid-water clouds and the observable cloud depolarization ratio profiles. The simulation model is introduced in Sect. 3.3. The development of the cloud retrieval scheme is outlined in Sect. 4 based on extensive simulation studies. In Sect. 5, the uncertainties in the retrieved cloud properties are discussed. Sect. 6 presents the lidar data

analysis regarding the aerosol properties (below the investigated cloud layer) obtained with the same lidar. Sect. 7, finally summarizes the entire cloud and aerosol data analysis procedures and provides a final table with all data analysis steps. After the detailed description of the methodology in this part 1, a dual-FOV polarization lidar setup is described in part 2 (Jimenez et al., 2020). This lidar performed continuous aerosol and cloud observations at Punta Arenas (53°S) in southern Chile at pristine marine conditions of the Southern Ocean within the framework of an 18-month field campaign. In part 2, two case studies are

discussed to demonstrate the potential of the new lidar approach to study aerosol-cloud interaction of liquid water clouds.

## 2   Multiple scattering lidar

In the beginning, we provide a brief overview of lidar applications in liquid-water cloud research. The use of lidar to derive cloud properties from measurements of multiple scattering contributions to the return signals has a long tradition. Strong forward scattering of incident laser photons occurs on the way up to the in-cloud backscatter region and on the way back to

the lidar (Mooradian et al., 1979). The multiple scattering (MS) effect depends on the geometrical and spectral characteristics of the lidar instrument and on the geometrical and microphysical properties of the cloud layers (Bissonnette et al., 1995; Chaikovskaya, 2008).

    Several models are available to simulate the MS contribution to the lidar return signal (e.g. Eloranta, 1998; Hogan, 2008; Wandinger, 1998; Katsev et al., 1997; Chaikovskaya and Zege, 2004; Donovan et al., 2015) and many attempts have been

undertaken to explored the potential of lidar to retrieve optical and microphysical properties of liquid-water clouds from measured multiple scattering effects (e.g. Pal and Carswell, 1985; Roy et al., 1999; Bissonnette et al., 2005; Kim et al., 2010; Schmidt et al., 2013; Donovan et al., 2015). A promising way is the use of a lidar measuring cloud backscatter signals at several FOVs. Bissonnette et al. (2005) proposed a multiple-FOV approach based on the measurement of total elastic-backscattering returns in combination with Monte-Carlo simulations. Roy et al. (1999) has introduced a robust approach based on cross-

polarized returns at multiple FOVs, allowing the assessment of the droplet size distribution.

    The information content in multiple-FOV polarization lidar returns was then systematically (theoretically and experimentally) studied by Veselovskii et al. (2006). This work demonstrated the ability of a multiple FOV lidar to investigate cloud microphysical properties in very large detail. One of the conclusions from this analysis is that the use of six FOVs would be





optimum and would allow an accurate retrieval of droplet sizes, amount, and light-extinction coefficient. However, the realization of a lidar receiver with six well-calibrated FOVs is challenging. Thus, in this study we propose a dual-FOV polarization lidar approach (in part 1) and demonstrate that such an attempt is easy to realize and provides high quality cloud measurements (in part 2). The sensitivity of such a dual-FOV lidar system to cloud microphysical properties depends on the selected pair of

FOVs and on the altitude of the target (Malinka and Zege, 2003; Veselovskii et al., 2006) as shown below.

Donovan et al. (2015) recently presented a new approach of a single-FOV polarization lidar-based method for the observation of liquid-water clouds. The retrieval is based on computed look-up tables of the cross- and co-polarized signal strength as a function of cloud microphysical properties. The cloud light-extinction coefficient and droplet effective radius can be retrieved by applying a Bayesian optimal estimation procedure. We will compare our results with the ones obtained with the method

suggested by Donovan et al. (2015).

## 3 Methodological background and cloud simulation model

In this section, we provide the theoretical background of the developed dual-FOV polarization lidar method. In Sect. 3.1, we begin with an overview of the retrievable cloud microphysical and observable optical properties of liquid-water clouds. Afterwards, we demonstrate how the measured volume linear depolarization ratio is related to the strength of multiple scattering

(MS) as a function of receiver FOV and given cloud properties (Sect. 3.2). This provides first insight into the relationship between light depolarization, cloud extinction, and droplet effective radius that we want to determine. Then we introduce the MS simulation model (Sect. 3.3) that was used to develop the dual-FOV polarization lidar technique (presented in Sect. 4 and 5) and show comparisons to demonstrate that the MS model is able to simulate real-world cloud scenarios, multiple scattering processes, and lidar backscatter signals.

### 3.1 Basic cloud microphysical and optical properties

As outlined and summarized by Schmidt et al. (2013, 2014) and Donovan et al. (2015), the basic properties characterizing a liquid-water cloud layer are the cloud droplet number concentration $N_d$, the cloud droplet effective radius $R_e$, the cloud droplet (single scattering) light-extinction coefficient $\alpha$ and the liquid-water content $w_l$. The liquid-water content of droplets in a given volume is defined as:

$$w_l = \frac{4}{3}\pi\rho_w \int_0^\infty n(r)r^3 dr = \frac{4}{3}\pi\rho_w \left( \frac{\int_0^\infty n(r)r^3 dr}{\int_0^\infty n(r)dr} \right) \int_0^\infty n(r)dr = \frac{4}{3}\pi\rho_w R_v^3 N_d \quad (1)$$

with the total droplet number concentration $N_d = \int_0^\infty n(r)dr$, the volume mean droplet radius $R_v$ of a given droplet size distribution, and the liquid-water density $\rho_w$.

The light-extinction coefficient of the cloud layer can be approximated by

$$\alpha = 2\pi \int_0^\infty n(r)r^2 dr = 2\pi R_s^2 N_d \quad (2)$$





in the case that the droplets are large in comparison to the laser wavelength. $R_s$ denotes the surface mean droplet radius. Besides the cloud extinction coefficient, the droplet effective radius

$$R_e = \frac{\int_0^\infty n(r) r^3 dr}{\int_0^\infty n(r) r^2 dr} = \frac{N_d R_v^3}{N_d R_s^2} \Rightarrow R_s^2 = \frac{R_v^3}{R_e} \tag{3}$$

is used to characterize an observed liquid-water cloud layer. By combining Eqs. (1), (2), and (3) we can write for the liquid-water content:

$$w_l = \frac{2}{3} \rho_w \alpha R_e . \tag{4}$$

Based on in-situ measurements in warm stratified clouds Martin et al. (1994) found that the cubic power of the measured

effective radius and the cubic power of the volume mean droplet radius follow a linear relationship, defining the parameter $k$:

$$k = \frac{R_v^3}{R_e^3} . \tag{5}$$

This linear relationship suggests that, in most of cases, a modified gamma function (Eq. (2) in Schmidt et al. (2014), Eq. (6) in Donovan et al. (2015)) can describe the droplet size distribution. Lu and Seinfeld (2006) compiled a list of $k$ values for stratiform clouds based on a literature review. The $k$ range of 0.75±0.15 well represents the values found for continental air

masses. For marine stratocumulus $k$ was slightly larger (around 0.8).

From Eqs. (2), (3), and (5) an expression for the cloud droplet number concentration can be obtained:

$$N_d = \frac{1}{2\pi k} \alpha R_e^{-2} . \tag{6}$$

Eqs. (4) and (6) permit the calculation of the liquid water content $w_l$ and the droplet number concentration $N_d$ from lidar measurements of the cloud extinction coefficient $\alpha$ and the droplet effective radius $R_e$ as already outlined by Schmidt et al.

(2013, 2014). In the next sections, we evaluate the possibilities of retrieving information about these two observable parameters from lidar measurements of cloud depolarization ratios caused by multiple scattering. The investigation is based on simulations with a semi-analytical scattering model (introduced in Sect. 3.3) which can compute the co- and cross-polarized lidar returns in multiple scattering regimes of pure liquid-water clouds.

### 3.2 Relationship between light depolarization and multiple scattering

It is well known that the polarization state of backscattered photons (at 180° scattering angle) remains invariant in cases of single scattering by spherical water droplets. In dense water clouds (multiple scattering regime), however, one or more forward scattering process take place so that the backscatter process, that allows the laser photons to return to the receiver telescope within the lidar FOV, occurs at a near 180° scattering angle. In this case of backscattering, rotations of the polarization plane of the laser pulse will occur (Zege and Chaikovskaya, 1996). This multiple scattering effect causes depolarization of the incident

linearly polarized laser light which can be described with Mie theory and can be physically understood from the angular scattering properties of single water spheres (Sassen and Petrilla, 1986).





To provide an easy-to-follow overview of the polarimetric behavior in lidar-relevant multiple scattering regimes, let us start with a single scattering event. This is illustrated in Fig. 1a. In this case of a single scattering of laser photons at scattering angle $\theta$, the Stokes vector $\mathbf{A}$ describing the resulting polarization state with respect to the receiver polarization axis can be obtained by multiplying the corresponding matrices that describe the coordinate transformation (Wandinger, 1994). The Stokes vector $\mathbf{A}$ for this single scattering event is given by

$$\mathbf{A}(\theta,\phi) = \mathbf{B}^{\mathrm{sc}\to\mathrm{cc}}(\theta,\phi)\mathbf{P}(\theta)\mathbf{B}^{\mathrm{cc}\to\mathrm{rc}}(\phi)\mathbf{A}_{\mathrm{lin}}. \tag{7}$$

The polarization state is defined with respect to the scattering plane indicated as colored region in Fig. 1a. The transformation matrix $\mathbf{B}^{\mathrm{cc}\to\mathrm{rc}}(\phi)$ enables the transition from the Cartesian coordinate system (cc, $\boldsymbol{e}_{\|\mathrm{cc}}$, $\boldsymbol{e}_{\perp\mathrm{cc}}$, $\boldsymbol{e}_{\mathrm{z}}$ coordinates in Fig. 1a) to the $\phi$-rotated system (rc, $\boldsymbol{e}_{\|\mathrm{rc}}$, $\boldsymbol{e}_{\perp\mathrm{rc}}$, $\boldsymbol{e}_{\mathrm{z}}$ coordinates) after scattering of the incident wave front. $\mathbf{P}$ represents the single scattering matrix defined for an isotropic media (Zege and Chaikovskaya, 2000) and $\mathbf{A}_{\mathrm{lin}}$ the Stokes vector for the 100% linearly polarized laser pulses, associated with the scattering plane (defined by $\boldsymbol{e}_{\|\mathrm{cc}}$, $\boldsymbol{e}_{\perp\mathrm{cc}}$, $\boldsymbol{e}_{\mathrm{z}}$, initial laser polarization plane). The transformation matrix $\mathbf{B}^{\mathrm{sc}\to\mathrm{cc}}$ enables finally the transition from the scattering-coordinate system (sc, $\boldsymbol{e}_{\|\mathrm{sc}}$, $\boldsymbol{e}_{\perp\mathrm{sc}}$, $\boldsymbol{e}_{\mathrm{r}}$ coordinates in Fig. 1a) to the original cartesian system (cc).

Next, we consider the double scattering event consisting of forward scattering of laser photons by one droplet at height $z_{\mathrm{f}}$ at a small forward scattering angle $\theta_{\mathrm{f}}$ and backward scattering by another droplet at height $z_{\mathrm{b}}$ at a wide angle $\theta_{\mathrm{b}} = \pi - \theta_{\mathrm{f}}$ (around $180°$). This is illustrated in Fig. 1b. The Stokes vector is now given by

$$\mathbf{A}(\theta_{\mathrm{f}},\theta_{\mathrm{b}},\phi) = \mathbf{B}^{\mathrm{sc}\to\mathrm{cc}}(\phi)\mathbf{P}(\theta_{\mathrm{b}})\mathbf{P}(\theta_{\mathrm{f}})\mathbf{B}^{\mathrm{cc}\to\mathrm{rc}}(\phi)\mathbf{A}_{\mathrm{lin}}. \tag{8}$$

Forward and backward scattering is separately described by the matrices $\mathbf{P}(\theta_{\mathrm{f}})$ and $\mathbf{P}(\theta_{\mathrm{b}})$. For this simple double scattering scenario, we have $\mathbf{B}^{\mathrm{sc}\to\mathrm{cc}}(\phi) = \mathbf{B}^{\mathrm{cc}\to\mathrm{rc}}(-\phi)$

From the Stokes vector $\mathbf{A}(\theta_{\mathrm{f}},\theta_{\mathrm{b}},\phi)$ we can compute the co and cross-polarized lidar signal components $S_{\|}$ and $S_{\perp}$. In Fig. 2b, the computed azimuthal pattern (in the backscatter plane orthogonal to the z-axis in Fig. 1b) of the co and cross-polarized signal components for scattering angles from 170 - $180°$ are shown for four different droplet sizes. Azimuthal patterns can not be observed with lidar systems. The lidar receiver will collect the scattered light over the entire azimuthal range and store it as one signal. However, by selecting a certain receiver FOV, we define the range of scattering angles $\theta_{\mathrm{f}}$ and $\theta_{\mathrm{b}}$ that a lidar can detect in the multiple scattering regime and by measuring lidar return signals at different FOVs and thus for different ranges of $\theta_{\mathrm{f}}$ and $\theta_{\mathrm{b}}$, a way is opened to derive information about the droplet sizes as emphasized in Fig. 2a-d. This is the basic idea of combining lidar measurements at different FOVs to retrieve the effective radius of the droplets and, in the next step, further cloud properties as will be described in Sect. 4.

From the two observed lidar return signal components, $S_{\perp}$ and $S_{\|}$ backscattered at height $z_{\mathrm{b}}$, the so-called volume or, in the case of dense water clouds, droplet linear depolarization ratio defined as

$$\delta(z_{\mathrm{b}}) = \frac{S_{\perp}(z_{\mathrm{b}})}{S_{\|}(z_{\mathrm{b}})} \tag{9}$$

is obtained. After forward scattering, the laser photons are backscattered at a certain backscatter angle $\theta_{\mathrm{b}}$. The dependence of the depolarization ratio on the backscatter angle $\theta_{\mathrm{b}}$ is shown in Fig. 2c for the four droplet effective radii. It can be seen that





the depolarization ratio increases to considerable values when the scattering angle deviates from $180°$. This sensitivity of the non-$180°$ backscattering angle $\theta_b$ on light depolarization and the strong forward scattering peak in Fig. 2a are the features used
in the dual-FOV polarization lidar technique to retrieve the basic cloud microphysical properties. Fig. 2a, c, and d provide an impression of the sensitive impact of cloud droplet size on measurable lidar quantities and thus suggest again that polarization lidars operated at two FOVs have the potential to derive $R_e$ and subsequently also the cloud extinction coefficient $\alpha$. Both, $R_e$ and $\alpha$ are closely linked to the cloud droplet number concentration $N_d$ (see Eq. 6). The relationship between MS-induced light depolarization measured at several FOVs and the cloud droplet size characteristics has already been illuminated and discussed
in previous studies (Veselovskii et al., 2006; Roy et al., 2016). In the next sections, we will show that a dual-FOV polarization lidar can already provide trustworthy information about the size and extinction coefficient in the cloud base region of liquid water clouds.

To emphasize the dominating impact of the receiver FOV on the measured multiple scattering effects let us, at the end of this subsection, compare the influence of the laser beam width and divergence, receiver telescope area, and the receiver field of
view on the observable cloud volume. In the case of a receiver FOV of 1 mrad, the lidar sees or observes a geometrical cross section (circular area in the horizontal plane at cloud base height $z_{bot}$) of about 0.8 m$^2$, 7 m$^2$, and 20 m$^2$ for a cloud with base height at 1, 3, and 5 km, respectively. The observable cross sections increase to about 3 m$^2$, 28 m$^2$, and 80 m$^2$ when using a 2 mrad FOV. In contrast, in case of a 30 cm receiver telescope (and a theoretical FOV of 0 mrad), the monitored circular cloud area at cloud base is less than 0.1 m$^2$. Also the divergence of the laser beam (0.1 to 0.2 mrad) has only a minor impact on the
amount of backscattered photons (and MS effects). The illuminated cloud cross section at cloud base is always $<1$ m$^2$ for a cloud base height of $<5$ km. So, the FOV clearly determines the cloud volume (geometrical cross section at cloud base times 50-100 m laser beam penetration depth into the cloud) available for MS cloud studies with lidar.

### 3.3  Multiple scattering model

After presenting the principle relationship between the measured linear depolarization ratio, forward scattering, and droplet
size, next we introduce the multiple scattering model used to develop our retrieval method presented in Sect. 4. The simulation model allows us to simulate realistic cloud scenarios with varying cloud height, droplet number concentration, cloud extinction coefficient, and droplet size distribution and the resulting co- and cross–polarized lidar signal components $S_\parallel$ and $S_\perp$ for a given lidar configuration parameters such as laser beam divergence and receiver FOV. The model is not restricted to single and doube scattering events. It simulates multiple forward scattering processes and one backscattering process. In several
articles, the radiative transfer problem of polarized light undergoing multiple scattering in an optically dense medium has been addressed (analytically) and several solutions have been proposed and tested (Zege et al., 1995; Zege and Chaikovskaya, 1999, 2000). The so-called small-angle approximation is adopted in this work. This is justified in the case of a narrow and pronounced forward scattering peak of the droplet scattering phase function which in turn is the case when the droplet size (of the order of 5 - 20 μm) is large compared to the laser wavelength (532 nm). This simplifying approach offers high accuracy together with high computing efficiency.



The Stokes vector $\mathbf{A}$ has the general form $\mathbf{A} = (I, Q, U, V)^{\mathrm{T}} = (S_{\parallel} + S_{\perp}, S_{\parallel} - S_{\perp}, U, V)^{\mathrm{T}}$ and the Stokes vector is $\mathbf{A}_{\mathrm{lin}} = (1, 1, 0, 0)^{\mathrm{T}}$ in the case of linearly polarized laser pulses (in x direction in Fig. 1) in Eqs. (7) and (8). The first element of the Stokes vector is the light intensity $I$ which satisfies the radiative transfer equation after multiplication with the single scattering

matrix element $P_{11}(\theta)$ (shown after normalization in Fig 2a). The $Q$ component of the Stokes vector describes the linear polarization and satisfies the same equation but with the 'modified' angular scattering function that equals $(P_{22} \pm P_{33})/2$ with the sign '+' for the forward and '-' for backward scattering. $P_{22}$ and $P_{33}$ are also elements of the scattering matrix $\mathbf{P}$. In this way, the Stokes vector components can be solved separately as a scalar radiative transfer problem (Zege and Chaikovskaya, 2000). The authors emphasized the potential of the model for developing new retrieval techniques. The scattering elements

required to initialize the multiple scattering model were calculated by using the ATMOTOOLS package (Zege et al. , 1993).

The modelled components $I$ and $Q$ enable the calculations of the cross- and co-polarized returns $S_{\perp}(z_{\mathrm{b}})$ and $S_{\parallel}(z_{\mathrm{b}})$ for backscatter height $z_{\mathrm{b}}$ within a liquid-water cloud layer,

$$I(\boldsymbol{X}(\boldsymbol{z}_{\mathbf{b}}), \boldsymbol{G}) = S_{\parallel}(z_{\mathrm{b}}) + S_{\perp}(z_{\mathrm{b}}), \qquad (10)$$

$$Q(\boldsymbol{X}(\boldsymbol{z}_{\mathbf{b}}), \boldsymbol{G}) = S_{\parallel}(z_{\mathrm{b}}) - S_{\perp}(z_{\mathrm{b}}). \qquad (11)$$

The volume depolarization ratio according to Eq. (9) is then given by

$$\boldsymbol{\delta}(z_{\mathrm{b}}) = \frac{I(z_{\mathrm{b}}) - Q(z_{\mathrm{b}})}{I(z_{\mathrm{b}}) + Q(z_{\mathrm{b}})}. \qquad (12)$$

The geometrical vector $\boldsymbol{G}(\boldsymbol{\Theta}_{\mathbf{ldiv}}, \boldsymbol{\Theta}_{\mathbf{fov}}, \boldsymbol{d}_{\mathbf{lb}}, \boldsymbol{d}_{\mathbf{m1}}, \boldsymbol{d}_{\mathbf{m2sd}})$ required to solve Eqs. (10) and (11) provides all necessary information about the lidar configuration in terms of the full divergence angle of the laser beam $\Theta_{\mathrm{ldiv}}$, the beam diameter $d_{\mathrm{lb}}$, the FOV full divergence angle of the receiver $\Theta_{\mathrm{fov}}$, the diameter of the primary receiver telescope $d_{\mathrm{m1}}$ and its respective second

mirror shadow $d_{\mathrm{m2sd}}$. The atmospheric state vector $\boldsymbol{X}(\boldsymbol{\alpha}(\boldsymbol{z}_{\mathbf{b}}), \boldsymbol{R}_{\mathbf{e}}(\boldsymbol{z}_{\mathbf{b}}))$ provides the cloud information, i.e., cloud extinction coefficient $\alpha(z_{\mathrm{b}})$ (assumed as the scattering coefficient) and effective radius $R_{\mathrm{e}}$ at height $z_{\mathrm{b}}$.

### 3.4 Model quality check: Comparison with ECSIM Monte-Carlo simulations and CALIPSO multiple scattering observations

We investigated to what extend the used MS model is able to simulate real-world polarization lidar observations and thus

can be used to develop new lidar analysis methods with focus on clouds. We compared our simulations with results obtained with the Monte-Carlo simulation model ECSIM (EarthCARE Simulator) (Donovan et al., 2015, 2010) and observations with the CALIPSO (Cloud–Aerosol Lidar and Infrared Pathfinder Satellite Observation) lidar (Hu et al., 2007). EarthCARE (Earth Clouds, Aerosols and Radiation Explorer) is a planned spaceborne lidar and radar mission, designed within a co-operation of the European Space Agency (ESA) and the Japan Aerospace Exploration Agency (JAXA) (Illingworth et al., 2015). Based

on the 4×4 $\alpha$-$R_{\mathrm{e}}$ combinations (4 $\alpha(z_{\mathrm{b}})$ values in the range from 5-26 km$^{-1}$, 4 $R_{\mathrm{e}}(z_{\mathrm{b}})$ values in the range from 3-15 $\mu$m), we performed more than 200 different simulations for these 16 cloud scenarios by considering cloud penetration depths from 10–70 m (with step width of 10 m), two different FOVs of 0.5 and 2.0 mrad, and assuming a liquid-cloud layer with cloud base height at 3000 m. We compared the obtained cloud-integrated volume depolarization ratio with respective values simulated with





the Monte-Carlo simulation model ECSIM in Fig. 3a. As can be seen, our simulations ($\delta$(our model)) are in good agreement with results of the sophisticated Monte-Carlo model. Both models agree for most of the integrated depolarization ratio, except

for the largest penetration depth causing volume depolarization ratio values close to 0.1. On average the depolarization ratios obtained with ECSIM are 0.016 larger than our values. The small differences indicate that our method delivers a realistic picture of multiple scattering in liquid-water clouds. The growing disagreement for depolarization ratios > 0.05 are caused by different assumptions and implementations regarding the considered narrow ranges of the small-angle forward scattering processes and the one wide angle backscattering process in the different models.

In a second approach, we compared our simulations with CALIPSO polarization lidar observations. Hu et al. (2007) investigated the relationship between the ratio of the total, cloud-integrated lidar return signal $\overline{\gamma}$ (from cloud top to base in the case of the CALIPSO lidar) to the one caused by single scattering ($\overline{\gamma}_{ss}$ caused by one backscattering process) and the respective cloud-integrated linear depolarization ratio $\overline{\delta}$. This study was based on observations with ground-based and spaceborne lidars supported by sophisticated Monte-Carlo simulations of the multiple scattering impact on the observed cloud lidar returns. By

performing a polynomial regression analysis to all observations they found the following best matching relationship

$$\frac{\overline{\gamma}}{\overline{\gamma}_{ss}} = \left( \frac{1 - \overline{\delta}}{1 + \overline{\delta}} \right)^2 . \tag{13}$$

The measured cloud-integrated CALIPSO lidar signal $\overline{\gamma}$ results from single plus multiple scattering events and correspond to the respective cloud-integrated depolarization ratio $\overline{\delta}$ for a given receiver FOV full angle $\Theta$.

In Fig. 3b, the relationship presented by Hu et al. (2007) is shown as a solid black line. As can be seen, our individual

simulations for the two FOVs (red and black circles) are in good agreement with Eq. (13) (black solid line in Fig. 3b) which again corroborates that our model well describes the link between cloud multiple scattering and light depolarization.

## 4   Retrieval of microphysical properties from polarization lidar observations at two FOVs

Based on simulations, the goal is to establish a method that allows us to retrieve $R_e$ and $\alpha$ from measured $\delta$ values at two FOVs, and afterwards to determine $w_l$ and $N_d$ by means of $R_e$ and $\alpha$. Therefore a large number of polarization lidar measurements for

the full range of observable parameters were simulated with the MS model and formed the basis for the development of the new dual-FOV lidar measurement and data analysis concept. The cloud parameters in Table 1 served as input in the simulations. The sketch in Fig. 4 provides an overview about the height profiles of the most relevant cloud parameters. The cloud is assumed to be in a subadiabatic equilibrium (Albrecht et al., 1990) in the lowest 100-200 m as typically given in liquid-water clouds (Merk et al., 2016; Foth and Pospichal, 2017; Barlakas et al., 2020). The same subadiabatic conditions are assumed in the

single-FOV polarization lidar approach of Donovan et al. (2015). Our data analysis scheme introduced below will deliver the cloud microphysical products for height $z_{ref}$ that is 50–100 m above cloud base height $z_{bot}$. The respective cloud penetration depth for laser light pulses is defined as

$$\Delta z_{ref} = z_{ref} - z_{bot} . \tag{14}$$





Following the methodological approach as outlined by Donovan et al. (2015), we assume that the cloud droplet number concentration $N_\mathrm{d}$ (Eq. 6) is height-independent and the liquid water content $w_\mathrm{l}(z)$ increases linearly with height (see Fig. 4). The profile of the liquid-water content (Eq. 4) can thus be expressed by

$$w_\mathrm{l}(z) = \Gamma_\mathrm{l} \Delta z \tag{15}$$

with the gradient of the liquid-water content $\Gamma_\mathrm{l} = dw_\mathrm{l}/dz$ for subadiabatic cloud conditions and the column depth

$$\Delta z = z - z_\mathrm{bot}. \tag{16}$$

Cloud droplets form at cloud base and then grow by water uptake at supersaturation conditions in updraft regions. According to Eqs. (4) and (15) we can write

$$\Gamma_\mathrm{l} \Delta z = \frac{2}{3} \rho_\mathrm{w} \alpha(z) R_\mathrm{e}(z). \tag{17}$$

By using Eqs. (6) and (17) and forming the ratio $\Gamma_\mathrm{l} \Delta z / N_\mathrm{d}$ we obtain

$$\frac{\Gamma_\mathrm{l} \Delta z}{N_\mathrm{d}} = \frac{4\pi k \rho_\mathrm{w}}{3} R_\mathrm{e}^3(z) \tag{18}$$

and for $R_\mathrm{e}(z)$

$$R_\mathrm{e}(z) = \left( \frac{3\Gamma_\mathrm{l} \Delta z}{4\pi \rho_\mathrm{w} k N_\mathrm{d}} \right)^{1/3}. \tag{19}$$

Further treatment leads to the link between $R_\mathrm{e}(z)$ and $R_\mathrm{e}(z_\mathrm{ref})$,

$$R_\mathrm{e}(z) = R_\mathrm{e}(z_\mathrm{ref}) \left( \frac{z - z_\mathrm{bot}}{z_\mathrm{ref} - z_\mathrm{bot}} \right)^{1/3}. \tag{20}$$

The $R_\mathrm{e}(z)$ profile is shown in Fig. 4.

To obtain the profile of the cloud extinction coefficient $\alpha(z)$ used in the simulations we combine Eqs. (17) and (19),

$$\Gamma_\mathrm{l} \Delta z = \frac{2}{3} \rho_\mathrm{w} \alpha(z) \left( \frac{3\Gamma_\mathrm{l} \Delta z}{4\pi \rho_\mathrm{w} k N_\mathrm{d}} \right)^{1/3}. \tag{21}$$

Rearrangement yields

$$\alpha(z) = \frac{3\Gamma_\mathrm{l} \Delta z}{\rho_\mathrm{w}} \left( \frac{\pi k N_\mathrm{d} \rho_\mathrm{w}}{2(3\Gamma_\mathrm{l} \Delta z)} \right)^{1/3}, \tag{22}$$

$$\alpha(z) = \left( \frac{3\Gamma_\mathrm{l} \Delta z}{\rho_\mathrm{w}} \right)^{2/3} \left( \frac{\pi k N_\mathrm{d}}{2} \right)^{1/3}. \tag{23}$$

Finally, we can write:

$$\alpha(z) = \alpha(z_\mathrm{ref}) \left( \frac{z - z_\mathrm{bot}}{z_\mathrm{ref} - z_\mathrm{bot}} \right)^{2/3}. \tag{24}$$





The profile of $\alpha(z)$ is shown in Fig. 4 as well.

Now we used the profiles in Fig. 4, described by Eqs. (6), (15), (19), and (23), to simulate the corresponding cross- and co-polarized lidar backscatter returns and the volume depolarization ratio (Eq. 12). We performed computations at two receiver

FOVs for 720 different cloud scenarios (defined by the state vector $\boldsymbol{X}$) by using Eqs. (10)-(11). The input parameters (10 values for $\alpha(z_{\mathrm{ref}})$, 9 values for $R_{\mathrm{e}}(z_{\mathrm{ref}})$, and 8 cloud base altitudes $z_{\mathrm{bot}}$) are given in Table 1. Overall cloud depth was 200 m. Vertical resolution or step width in the computations was 7.5 m which corresponds to the vertical resolution of the lidar observations introduced in part 2 (Jimenez et al., 2020).

Fig. 5 shows the profiles of the linear depolarization ratios $\delta_{\mathrm{in}}$ for the inner and outer FOVs, i.e., for FOV$_{\mathrm{in}}$ of 1 mrad

and for FOV $\delta_{\mathrm{out}}$ of 2 mrad for four different profiles of the cloud extinction coefficient $\alpha$ and four different profiles of the effective radius $R_{\mathrm{e}}$ of the droplets. The simulated cloud layer is at 3 km height. A monotonic increase of the volume linear depolarization ratio is visible because of the increasing contribution of multiple scattering processes to the amount of backscattered laser photons with increasing cloud penetration depths. With increasing number of cloud droplets and thus increasing light extinction the probability of multiple scattering strongly increases and thus the strength of depolarization.

The striking feature in Fig. 5 is the clear dependence of the droplet effective radius $R_{\mathrm{e}}(z_{\mathrm{ref}})$ on $\delta_{\mathrm{in}}/\delta_{\mathrm{out}}$. In principle, we can show a similar figure by combining different backscatter signals measured with lidar at two different FOVs. However, the comparison of all these combinations clearly revealed that the optimum retrieval of the cloud effective radius (as shown in Fig. 5c) is only possible by means of the co- and cross-polarized signal components observed at different FOVs.

According to Fig. 5c it is recommended to use the lidar observations in the lowest part of the liquid-water cloud to retrieve

the cloud microphysical properties. To obtain robust values of the cloud depolarization ratios at the two different FOVs (with low signal noise impact) we integrate, in the next step, the depolarization ratio from the cloud base to a fixed reference altitude (see Fig. 4):

$$\overline{\delta}_{\mathrm{in}}(z_{\mathrm{bot}}, z_{\mathrm{ref}}) = \frac{\int_{z_{\mathrm{bot}}}^{z_{\mathrm{ref}}} S_{\perp,\mathrm{in}}(z)dz}{\int_{z_{\mathrm{bot}}}^{z_{\mathrm{ref}}} S_{\|,\mathrm{in}}(z)dz}, \tag{25}$$

$$\overline{\delta}_{\mathrm{out}}(z_{\mathrm{bot}}, z_{\mathrm{ref}}) = \frac{\int_{z_{\mathrm{bot}}}^{z_{\mathrm{ref}}} S_{\perp,\mathrm{out}}(z)dz}{\int_{z_{\mathrm{bot}}}^{z_{\mathrm{ref}}} S_{\|,\mathrm{out}}(z)dz}, \tag{26}$$

and further define the dual-FOV ratio of depolarization ratios,

$$\overline{\delta}_{\mathrm{rat}}(z_{\mathrm{bot}}, z_{\mathrm{ref}}) = \frac{\overline{\delta}_{\mathrm{in}}(z_{\mathrm{bot}}, z_{\mathrm{ref}})}{\overline{\delta}_{\mathrm{out}}(z_{\mathrm{bot}}, z_{\mathrm{ref}})}. \tag{27}$$

Malinka and Zege (2007) presented a method to check the sensitivity of the dual-FOV retrieval method to the selected pair of FOVs. We performed simulations with FOVs from 0.5–3.0 mrad and found that the highest sensitivity (optimum pair of

FOVs) is given for the case with the highest FOV$_{\mathrm{out}}$-to-FOV$_{\mathrm{in}}$ ratio. However, the selection of FOV$_{\mathrm{in}}$ of 1 mrad and FOV$_{\mathrm{out}}$ of 2 mrad as used in the following was found to be sufficiently sensitive for liquid-water cloud studies and, on the other hand, a good compromise when keeping cloud inhomogeneities into consideration. This topic will be discussed in part 2. The





backscatter signals may be different for the two FOVs not only because of the different multiple scattering contributions, but also because of the differences in the amount of photons backscatter from different cloud cross sections and cloud volumes (defined by cloud height and FOV) as a result of inhomogeneities in the cloud droplet number concentration that may vary in the horizontal plane.

In Fig. 6, an overview of all simulations of $\overline{\delta}_{\mathrm{rat}}$ and $\overline{\delta}_{\mathrm{in}}$ for 90 cloud scenarios (all possible combinations of cloud extinction and effective radii in Table 1) are shown for a cloud layer at $z_{\mathrm{bot}}$=3 km and FOVs of 1 and 2 mrad. As mentioned, the depolarization ratio values are integrated over the lowest 75 m of the cloud layer. Again, a clear dependence of $\overline{\delta}_{\mathrm{rat}}$ on the effective radius $R_{\mathrm{e}}$ at $z_{\mathrm{ref}}$ (75 m above cloud base) is visible in Fig. 6a. The dominating impact of the cloud extinction coefficient on $\overline{\delta}_{\mathrm{in}}$ is shown in Fig. 6b.

In Fig. 7, the relationship between $\overline{\delta}_{\mathrm{rat}}$ and effective radius $R_{\mathrm{e}}(z_{\mathrm{ref}})$ is presented for all cloud layers with base heights from 1 to 5 km and FOVs of 1 and 2 mrad. The horizontal bars indicate the influence of the cloud extinction coefficient for each of the simulated nine effective radii for the eight cloud layers. A polynomial regression is applied to each of the eight cloud simulation data sets and the respective cubic polynomial fits (Eq. 28) are shown as colored curves in Fig. 7.

     Eq. (28) is now used in our dual-FOV method to derive the droplet effective radius $R_{\mathrm{e}}(z_{\mathrm{ref}})$ from the measurements of $\overline{\delta}_{\mathrm{rat}}$

for the integration length $\Delta z_{\mathrm{ref}}$=75 m:

$$R_{\mathrm{e}}(z_{\mathrm{ref}}) = R_0 + R_1 \times \overline{\delta}_{\mathrm{rat}} + R_2 \times \overline{\delta}_{\mathrm{rat}}^2 + R_3 \times \overline{\delta}_{\mathrm{rat}}^3 . \tag{28}$$

The polynomial coefficients $R_0$, $R_1$, $R_2$, and $R_3$ are given in Table 2. For a given cloud base altitude, we obtained the appropriate curve by interpolating the two nearest curves (computed by means of the Table 2 values) for the adjacent heights.

     In the second step of the retrieval, the cloud extinction coefficient $\alpha(z_{\mathrm{ref}})$ is determined by using the derived effective radius

and the measured integrated depolarization ratio $\overline{\delta}_{\mathrm{in}}$ inserted in the quadratic polynomial fit,

$$\alpha(z_{\mathrm{ref}}) = \alpha_0(R_{\mathrm{e}}, z_{\mathrm{bot}}) + \alpha_1(R_{\mathrm{e}}, z_{\mathrm{bot}}) \times \overline{\delta}_{\mathrm{in}} + \alpha_2(R_{\mathrm{e}}, z_{\mathrm{bot}}) \times \overline{\delta}_{\mathrm{in}}^2 . \tag{29}$$

The coefficients $\alpha_0(R_{\mathrm{e}}, z_{\mathrm{bot}})$, $\alpha_1(R_{\mathrm{e}}, z_{\mathrm{bot}})$, and $\alpha_2(R_{\mathrm{e}}, z_{\mathrm{bot}})$ are obtained from a polynomial regression analysis applied to each simulation data set for a given cloud layer characterized by $z_{\mathrm{bot}}$ and $R_{\mathrm{e}}(z_{\mathrm{ref}})$ as well as the given inner FOV. Fig. 7b shows the relationship between the different parameters. The large data set of $\alpha$ coefficients are stored as look up tables and

are not presented in this paper.

     The two-step retrieval is finally explained again in Fig. 8 for a cloud layer with cloud base height of 3 km, $z_{\mathrm{ref}}$ at 75 m above cloud base height, and FOVs of 1 and 2 mrad. To show again the low dependency of $\overline{\delta}_{\mathrm{rat}}$ on cloud extinction, all ten simulations with $\alpha(z_{\mathrm{ref}})$ values from 5.2–28.6-km$^{-1}$ are presented. A clear relationship between $\overline{\delta}_{\mathrm{rat}}$ and $R_{\mathrm{e}}(z_{\mathrm{ref}})$ according to Eq. (28) is given. Fig. 8b is the basis for the second step of the retrieval. Here, the polynomial fits (Eq. 29) of the $\alpha(z_{\mathrm{ref}})$-

vs-$\overline{\delta}_{\mathrm{in}}$ simulations are used, and shown in Fig. 8b for the nine discrete effective radius values in Table 1. Thus, to avoid large errors in the $\alpha(z_{\mathrm{ref}})$ retrieval, $R_{\mathrm{e}}(z_{\mathrm{ref}})$ from step 1 is used to select the right curve for the $\alpha(z_{\mathrm{ref}})$ determination.

     Finally, after the derivation of the droplet extinction coefficient $\alpha(z_{\mathrm{Ref}})$ and the droplet effective radius $R_{\mathrm{e}}(z_{\mathrm{ref}})$ as independent variables, we can compute the liquid-water content $w_{\mathrm{l}}(z_{\mathrm{ref}})$ with Eq. (4) and the droplet number concentration $N_{\mathrm{d}}(z_{\mathrm{ref}})$ with Eq. (6), in the same way as presented by Schmidt et al. (2013, 2014).





## 5 Retrieval uncertainties

The retrieval of the effective radius $R_{\mathrm{e}}(z_{\mathrm{ref}})$ of the cloud droplets needs the ratio of depolarization ratios $\overline{\delta}_{\mathrm{rat}}$ and the cloud base height $z_{\mathrm{bot}}$ as input. The relationship between $R_{\mathrm{e}}(z_{\mathrm{ref}})$ and $\overline{\delta}_{\mathrm{rat}}$ is also a function of the cloud extinction coefficient $\alpha(z_{\mathrm{ref}})$. We can estimate the uncertainties caused by the uncertainty $\pm\Delta\overline{\delta}_{\mathrm{rat}}$ in the $\overline{\delta}_{\mathrm{rat}}$ measurement by calculating

$$\pm\sigma_{\mathrm{ran},R_{\mathrm{e}}}(\Delta\overline{\delta}_{\mathrm{rat}}) = R_{\mathrm{e}} \pm \left( R_0 + R_1 \times (\overline{\delta}_{\mathrm{rat}} \pm \Delta\overline{\delta}_{\mathrm{rat}}) + R_2 \times (\overline{\delta}_{\mathrm{rat}} \pm \Delta\overline{\delta}_{\mathrm{rat}})^2 + R_3 \times (\overline{\delta}_{\mathrm{rat}} \pm \Delta\overline{\delta}_{\mathrm{rat}})^3 \right) \tag{30}$$

and by taking half of the respective uncertainty bars.

Systematic retrieval uncertainties $\sigma_{\mathrm{sys},R_{\mathrm{e}}}$ arise from the use of the model (polynomial functions in Fig. 7), from the uncertainties in the determined cloud base height $\Delta z_{\mathrm{bot}}$ (we assume $\pm 15$ m), and the influence of the cloud extinction coefficient (the uncertainty is denoted here as $\Delta\alpha$ and given by the range of values in Table 1 from 5.2 to 28.6 Mm$^{-1}$). From the extended error simulations and from the analysis with real (observational) data we conclude that

we conclude that

$$\sigma_{\mathrm{sys},R_{\mathrm{e}}}(\Delta\alpha) \approx 0.15 R_{\mathrm{e}}(z_{\mathrm{ref}}), \tag{31}$$

$$\sigma_{\mathrm{sys},R_{\mathrm{e}}}(\Delta z_{\mathrm{bot}}) \approx 0.10 R_{\mathrm{e}}(z_{\mathrm{ref}}). \tag{32}$$

On average, input uncertainties may partly cancel out and the mean uncertainty is given by

$$\sigma_{\mathrm{sys},R_{\mathrm{e}}}(\Delta\alpha, \Delta z_{\mathrm{bot}}) = \sqrt{\sigma_{\mathrm{sys},R_{\mathrm{e}}}(\Delta\alpha)^2 + \sigma_{\mathrm{sys},R_{\mathrm{e}}}(\Delta z_{\mathrm{bot}})^2}. \tag{33}$$

The influence of measurement uncertainties on the retrieval of $\alpha(z_{\mathrm{ref}})$ is estimated by considering the standard deviation
$\pm\Delta\overline{\delta}_{\mathrm{in}}$ in the computation,

$$\pm\sigma_{\mathrm{ran},\alpha}(\Delta\overline{\delta}_{\mathrm{rat}}) = \alpha \pm \left( \alpha_0 + \alpha_1 \times (\overline{\delta}_{\mathrm{in}} \pm \Delta\overline{\delta}_{\mathrm{in}}) + \alpha_2 \times (\overline{\delta}_{\mathrm{in}} \pm \Delta\overline{\delta}_{\mathrm{in}})^2 \right). \tag{34}$$

In a similar way as described above for the systematic uncertainty in $R_{\mathrm{e}}$, we estimated $\sigma_{\mathrm{sys},\alpha}$ with $\Delta z_{\mathrm{bot}} \pm 15$ m and by using $\Delta R_{\mathrm{e}}$ according to Eq. (33) in the second retrieval step to obtain $\alpha(z_{\mathrm{ref}})$. Again, from many simulations we concluded that

$$\sigma_{\mathrm{sys},\alpha}(\Delta R_{\mathrm{e}}) \approx 0.08\alpha(z_{\mathrm{ref}}), \tag{35}$$

$$\sigma_{\mathrm{sys},\alpha}(\Delta z_{\mathrm{bot}}) \approx 0.15\alpha(z_{\mathrm{ref}}). \tag{36}$$

The overall mean systematic uncertainty may be given by:

$$\sigma_{\mathrm{sys},\alpha}(\Delta R_{\mathrm{e}}, \Delta z_{\mathrm{bot}}) = \sqrt{\sigma_{\mathrm{sys},\alpha}(\Delta R_{\mathrm{e}})^2 + \sigma_{\mathrm{sys},\alpha}(\Delta z_{\mathrm{bot}})^2}. \tag{37}$$

## 6 Retrieval of cloud-relevant aerosol properties and aerosol-cloud-interaction parameters

### 6.1 Lidar-derived aerosol properties

For completeness of the theoretical part 1, we briefly introduce the aerosol parameters needed for the ACI studies. Examples of aerosol observations with the Polly (*PO*ortab*L*e *L*idar s*Y*stem) instrument (Engelmann et al., 2016) used in part 2 and





upgraded to a dual-FOV polarization lidar can be found in Baars et al. (2016) and Hofer et al. (2017, 2020). The sketch in
Fig. 4 illustrates our overall concept of lidar-based ACI studies. The aerosol parameters are measured with the lidar at the
smaller FOV (FOV$_\text{in}$ several 100 m below cloud base. The cloud- and ACI-relevant aerosol proxies are the particle extinction
coefficient $\alpha_\text{par}(z)$ and the cloud condensation nucleus concentration $N_\text{CCN}(z)$. The methodology to derive $N_\text{CCN}$ profiles
from measurements of particle optical properties is outlined in Mamouri and Ansmann (2016). A brief summary of the method,

denoted as POLIPHON (Polarization Lidar Photometer Networking) method, is given here.

A specific problem in ACI studies is the retrieval of the particle backscatter and extinction profiles below extended liquid-
water cloud layers in the first step. The required calibration of the lidar profiles in clear air (at pure Rayleigh scattering
conditions), i.e., in the upper troposphere and lower stratosphere is then not possible. In these cases with aerosol backscatter
signals up to cloud base height $z_\text{bot}$ only, the so-called lidar constant is required in the retrieval of aerosol properties. The

determination of the lidar constant (considering all instrumental constants, such as laser pulse energy and receiver telescope
area, in the basic lidar equation) following the procedure of (Wiegner and Geiß, 2012) is performed during cloud-free situations
before or after the passage of the cloud fields or during periods with cloud holes so that clear air layers (Rayleigh scattering
regime) are available for the lidar calibration. Subsequently, the determined lidar constant is used during the cloudy periods
in the data analysis to retrieve the backscatter coefficient profiles up to the base of the optically dense water clouds again

following the procedure of Wiegner and Geiß (2012).

By means of height profiles of the aerosol particle depolarization ratio and the particle backscatter coefficient , the POLIPHON
data analysis separates particle backscatter and extinction contribution of the three basic aerosol types (marine aerosol, mineral
dust, anthropogenic haze). The aerosol-type-dependent 532 nm extinction coefficients below cloud base $z_\text{bot}$ are then converted
into particle number concentrations and respective CCN concentrations (for a water supersaturation level of 0.2% or relative

humidity over water of 100.2%) as described by Mamouri and Ansmann (2016).

For pure marine conditions, we obtain $N_\text{CCN}$ from $\alpha_\text{par}$ by using the following conversion:

$$N_\text{CCN} = 7(\alpha_\text{par})^{0.85} \tag{38}$$

with $N_\text{CCN}$ in cm$^{-3}$ and $\alpha_\text{par}$ im Mm$^{-1}$. For urban haze condition (central European pollution conditions), we apply:

$$N_\text{CCN} = 25(\alpha_\text{par})^{0.95} \tag{39}$$

and for desert dust

$$N_\text{CCN} = 4(\alpha_\text{par})^{0.9}. \tag{40}$$

The $N_\text{CCN}$ values assume that all dry particles with radius >50 nm (marine, urban) and >100 nm are potential cloud
condensation nuclei. The parameterization hold for an ambient relative humidity of 60% relative humidity for continental fine
mode aerosol and 80% relative humidity in the case of marine particles. Respective water-uptake effects by aerosol particles are
considered and corrected in Eqs. (38) and (39). In the case of hydrophobic dust particles, no water uptake effect is considered
and corrected.





The uncertainty in the basic aerosol-type-dependent extinction coefficients and in the retrieved $N_{\mathrm{CCN}}$ values is on the order
of 20% and 50-100%, respectively. However, aircraft comparisons (Düsing et al., 2018; Haarig et al., 2019) and long-term field
studies at a central European background station (Schmale et al., 2018) revealed that the uncertainty is typically of the order of
50%. It should be emphasized at the end that the Raman lidar Polly permits the retrieval of profiles of the water-vapor mixing
ratio and relative humidity (RH) (Dai et al., 2018) so that, in principle, actual RH measurements are available for the required
aerosol water uptake effects in the $N_{\mathrm{CCN}}$ conversion procedure as described by Mamouri and Ansmann (2016)

## 6.2   Aerosol-cloud-interaction (ACI) parameter

The study of the influence of aerosol particles on liquid-water cloud evolution and cloud microphysical properties is based on
two ACI parameters defined as (Feingold et al., 2001; McComiskey et al., 2009; Schmidt et al., 2013, 2014):

$$E_{\mathrm{ACI},\alpha_{\mathrm{par}}}(N_{\mathrm{d}},\alpha_{\mathrm{par}}) = d\ln(N_{\mathrm{d}})/d\ln(\alpha_{\mathrm{par}}) \tag{41}$$

and

$$E_{\mathrm{ACI},N_{\mathrm{CCN}}}(N_{\mathrm{d}},N_{\mathrm{CCN}}) = d\ln(N_{\mathrm{d}})/d\ln(N_{\mathrm{CCN}}). \tag{42}$$

The so–called nucleation–efficiency parameter $E_{\mathrm{ACI},\alpha_{\mathrm{par}}}$ describes the relative change of the cloud droplet number concentra-
tion $N_{\mathrm{d}}$ with a relative change in the particle extinction coefficient $\alpha_{\mathrm{par}}$. Correspondingly, $E_{\mathrm{ACI},N_{\mathrm{CCN}}}$ characterizes the relative
increase of $N_{\mathrm{d}}$ with a relative increase of the cloud condensation nucleus concentration $N_{\mathrm{CCN}}$. The higher the ACI value is the
stronger is the impact of the observed aerosol conditions on the cloud microphysical properties.

## 7   Summary

We presented a new polarization-based lidar approach to derive microphysical properties of pure liquid-water clouds. Extended
simulations with a MS model were performed regarding the relationship between cloud microphysical and light-extinction
properties and the cloud depolarization ratio measured with lidar at two different FOVs. These simulations served as the basis
for the development of the new dual-FOV polarization lidar method. An extended error analysis was performed as well. The
new dual-FOV polarization lidar technique can be combined with the POLIPHON method that allows the profiling of CCN
concentrations below cloud base. In Table 3, the full data analysis scheme of the dual-FOV polarization lidar is shown. All
steps of the data analysis procedure from the determination of the cloud microphysical properties and the aerosol proxies to
the ACI parameters are listed.

In part 2 (Jimenez et al., 2020), we describe how we implemented the novel dual-FOV polarization lidar technique in a
Polly instrument which is now used in a long-term field campaign in Punta Arenas, southern Chile, at the southern most tip
of South America. We present two case studies of this campaign in Part 2. Case 1 is used to explain the full aerosol and
cloud data analysis scheme in all details. This case study includes an uncertainty discussion and comparisons with alternative
approaches to derive cloud microphysical properties as the single-FOV polarization lidar technique (Donovan et al., 2015).



Based on case 2, the potential of the new lidar technique to improve significantly ACI studies in the case of liquid-water clouds is highlighted. The field site of Punta Arenas is surrounded by the Southern Ocean. Pristine marine conditions prevail. Continental and especially anthropogenic aerosol sources usually play a negligible role regarding their influence on cloud evolution and properties in this region of the world.

*Data availability.* All the analysis and simulation products are available at TROPOS upon request (info@tropos.de).

*Author contributions.* CJ and AA prepared the manuscript. CJ developed the method and performed all the simulations. AM provided the multiple scattering code. AA, RE, DD, AM, JS, and UW contributed to the design of the simulations study and to the discussion of the results.

CJ and AA prepared the manuscript. CJ developed the method and performed all the simulations. AM provided the multiple scattering code. AA, RE, DD, AM, JS, and UW contributed to the design of the simulations study and to the discussion of the results.

*Competing interests.* The authors declare that they have no conflict of interest.

*Special issue statement.* This article is part of the special issue "EARLINET aerosol profiling: contributions to atmospheric and climate research".

*Acknowledgements.* The authors like to thank the TROPOS lidar team members for their support and numerous fruitful discussions during the last five years.

*Financial support.* This research was partially funded by the program DAAD/Becas Chile, grant no. 57144001. This activity is supported by the ACTRIS Research Infrastructure (EU H2020-R&I), grant agreement no. 654109.





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





**Table 1.** Lidar and liquid-water cloud input parameters used in the simulations with the MS model.

| | |
|---|---|
| FOV, full solid angle, $\Theta_{\text{in}}$ (mrad) | 0.5, 1.0. 1.5, 2.0, 2.5 |
| FOV, full solid angle, $\Theta_{\text{out}}$ (mrad) | 1.0, 1.5, 2.0, 2.5, 3.0 |
| Cloud base height (km) | 1.0, 1.5, 2.0, 2.5, 3.0, 3.5, 4.0, 5.0 |
| $\alpha(\Delta z_{\text{ref}})$ (km$^{-1}$) for $\Delta z_{\text{ref}} = 75$ m | 5.2, 7.8, 10.4, 13.0, 15.6, 18.2, 20.8, 23.4, 26.0, 28.6 |
| $R_{\text{e}}(\Delta z_{\text{ref}})$ (μm) for $\Delta z_{\text{ref}} = 75$ m | 3.6, 4.7, 5.8, 6.9, 7.9, 9.4, 10.8, 12.6, 14.4 |





**Table 2.** Polynomial coefficients used in the computation of $R_e$ with Eq. (28).

| Height (km) | 1.0 | 1.5 | 2.0 | 2.5 | 3.0 | 3.5 | 4.0 | 5.0 |
|---|---|---|---|---|---|---|---|---|
| $\Theta_{in} = 0.5$ mrad, $\Theta_{out} = 2.0$ mrad | | | | | | | | |
| $R_3$ | 2786.1 | 645.26 | 260.76 | 147.93 | 104.62 | 81.783 | 68.905 | 55.418 |
| $R_2$ | -3051.8 | -816.25 | -364.36 | -218.56 | -158.59 | -124.77 | -104.21 | -79.808 |
| $R_1$ | 1165.6 | 380.43 | 199.36 | 133.71 | 103.87 | 85.513 | 73.159 | 56.151 |
| $R_0$ | -144.08 | -53.841 | -30.988 | -22.131 | -17.946 | -15.249 | -13.299 | -10.225 |
| Limits $\delta_{rat}$ | 0.275 | 0.289 | 0.308 | 0.329 | 0.348 | 0.368 | 0.387 | 0.422 |
| | $-0.433$ | $-0.530$ | $-0.616$ | $-0.685$ | $-0.738$ | $-0.780$ | $-0.812$ | $-0.859$ |
| $\Theta_{in} = 0.5$ mrad, $\Theta_{out} = 3.0$ mrad | | | | | | | | |
| $R_3$ | 818.53 | 242.71 | 118.52 | 77.835 | 61.364 | 51.452 | 46.136 | 40.688 |
| $R_2$ | -805.54 | -283.51 | -153.14 | -106.11 | -85.976 | -72.386 | -64.283 | -52.957 |
| $R_1$ | 306.27 | 141.59 | 92.215 | 71.816 | 62.092 | 54.779 | 49.727 | 40.714 |
| $R_0$ | -32.186 | -16.75 | -11.906 | -10.006 | -9.2678 | -8.6546 | -8.2073 | -6.9601 |
| Limits $\delta_{rat}$ | 0.203 | 0.226 | 0.252 | 0.279 | 0.306 | 0.331 | 0.355 | 0.399 |
| | $-0.413$ | $-0.524$ | $-0.616$ | $-0.686$ | $-0.739$ | $-0.778$ | $-0.808$ | $-0.849$ |
| $\Theta_{in} = 1.0$ mrad, $\Theta_{out} = 2.0$ mrad | | | | | | | | |
| $R_3$ | -313.06 | 14.365 | 103.89 | 191.77 | 321.68 | 484.96 | 711.03 | 1388.8 |
| $R_2$ | 610.34 | 7.5644 | -177.54 | -376.56 | -683.86 | -1082.2 | -1646.1 | -3386.1 |
| $R_1$ | -342.56 | 5.4226 | 122.17 | 264.95 | 500.91 | 819.29 | 1282.9 | 2761.7 |
| $R_0$ | 60.101 | -4.3288 | -27.946 | -61.543 | -121.32 | -205.48 | -331.71 | -748.7 |
| Limits $\delta_{rat}$ | 0.530 | 0.556 | 0.584 | 0.613 | 0.640 | 0.667 | 0.692 | 0.737 |
| | $-0.747$ | $-0.845$ | $-0.906$ | $-0.944$ | $-0.964$ | $-0.976$ | $-0.983$ | $-0.991$ |
| $\Theta_{in} = 1.0$ mrad, $\Theta_{out} = 3.0$ mrad | | | | | | | | |
| $R_3$ | -12.862 | 18.332 | 43.976 | 77.559 | 128.61 | 195.68 | 287.74 | 615.33 |
| $R_2$ | 20.399 | -17.111 | -61.857 | -129.4 | -239.1 | -388.14 | -599.27 | -1377.6 |
| $R_1$ | 23.446 | 25.775 | 45.587 | 86.695 | 161.91 | 269.5 | 428.24 | 1039.2 |
| $R_0$ | -8.03 | -5.9015 | -8.8799 | -17.449 | -34.762 | -60.691 | -100.41 | -259.75 |
| Limits $\delta_{rat}$ | 0.392 | 0.435 | 0.479 | 0.522 | 0.563 | 0.600 | 0.635 | 0.695 |
| | $-0.713$ | $-0.836$ | $-0.906$ | $-0.945$ | $-0.965$ | $-0.974$ | $-0.978$ | $-0.978$ |





**Table 3.** Overview of the cloud and aerosol retrieval procedure (step-by-step data analysis). The data analysis starts with a precise determination of the cloud base height $z_{\mathrm{bot}}$. The cloud products are given at the reference height $z_{\mathrm{ref}}$, 75 m above cloud base height $z_{\mathrm{bot}}$. In the estimation of the ACI efficiency, particle extinction and cloud condensation nucleus concentration at $z_{\mathrm{aer}}$, usually several 100 m below cloud base are considered.

| Parameter | Symbol | Equation | Uncertainty |
|---|---|---|---|
| Cloud base height | $z_{\mathrm{bot}}$ | | 0.1-1% |
| Cloud depolarization ratios | $\overline{\delta}_{\mathrm{in}}(z_{\mathrm{bot}}, z_{\mathrm{ref}})$ | Eq. (25) | 5% |
| | $\overline{\delta}_{\mathrm{out}}(z_{\mathrm{bot}}, z_{\mathrm{ref}})$ | Eq. (26) | 5% |
| | $\overline{\delta}_{\mathrm{rat}}(z_{\mathrm{bot}}, z_{\mathrm{ref}})$ | Eq. (27) | 10-15% |
| Droplet effective radius | $R_{\mathrm{e}}(z_{\mathrm{ref}})$ | Eq. (28) | 15% |
| Cloud extinction coefficient | $\alpha(z_{\mathrm{ref}})$ | Eq. (29) | 15-20% |
| Liquid water content | $w_{\mathrm{l}}(z_{\mathrm{ref}})$ | Eq. (4) | 25% |
| Cloud droplet number concentration | $N_{\mathrm{d}}(z_{\mathrm{ref}})$ | Eq. (6) | 25-75% |
| Aerosol depolarization ratio | $\delta_{\mathrm{par}}(z)$ | | 5-10% |
| Aerosol extinction coefficient | $\alpha_{\mathrm{par}}(z_{\mathrm{aer}})$ | | 20% |
| Cloud condensation nucleus concentration | $N_{\mathrm{CCN}}(z_{\mathrm{aer}})$ | Eq. (38) – (40) | 30-100% |
| Aerosol-cloud-interaction efficiency | $E_{\mathrm{ACI},\alpha_{\mathrm{par}}}(N_{\mathrm{d}}, \alpha_{\mathrm{par}})$ | Eq. (41) | |
| Aerosol-cloud-interaction efficiency | $E_{\mathrm{ACI},N_{\mathrm{CCN}}}(N_{\mathrm{d}}, N_{\mathrm{CCN}})$ | Eq. (42) | |

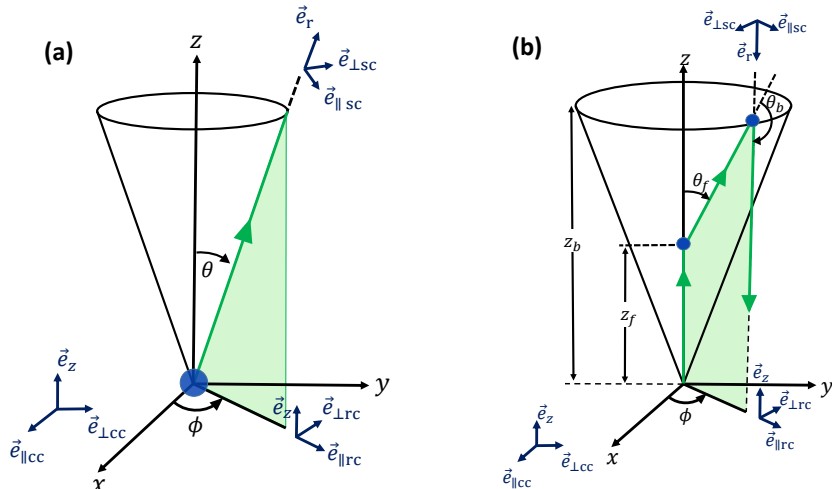

**Figure 1.** Scattering geometry for (a) one single scattering event and (b) one forward and one backward scattering event.



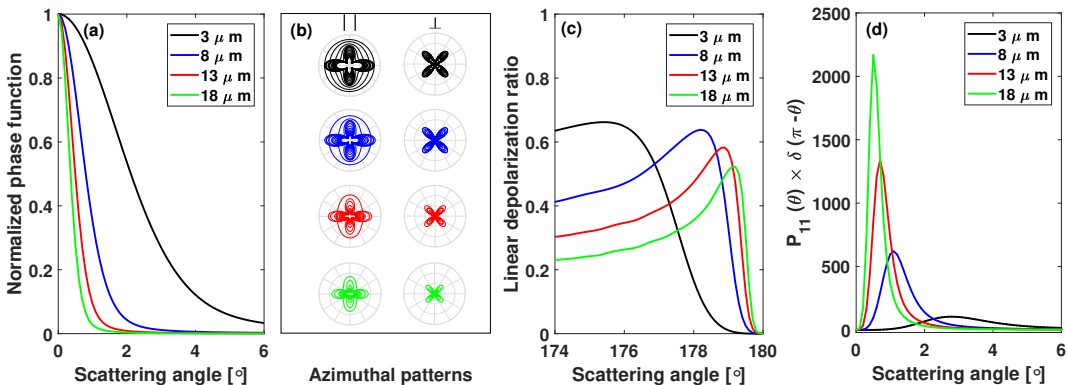

**Figure 2.** (a) Normalized scattering matrix element $P_{11}$ (normalized to the maximum at $0°$ scattering angle) as a function of forward scattering angle $\theta_f$ for four droplet diameters (given as numbers), (b) azimuthal patterns (computed with the MS model for the entire range of azimuthal angles from 0-$2\pi$, see Fig. 1b) of the co-polarized $\parallel$ and the cross-polarized $\perp$ signal components at scattering angles $\theta_b$ between 170 and $189.5°$ for the different droplet diameters, (c) droplet linear depolarization ratio $\delta = S_\perp/S_\parallel$ with the lidar signal components $S_\perp$ and $S_\parallel$ (obtained from azimuthal integration over the range from $\phi = 0 - 2\pi$ in b) as a function of the backscattering angle $\theta_b$ from $174°$ to $180°$ for the four droplet sizes, and (d) scattering matrix element $P_{11}(\theta)$ at $\theta_f$ (in a) multiplied by the depolarization ratio at $\theta_b = \pi - \theta$ (in c).

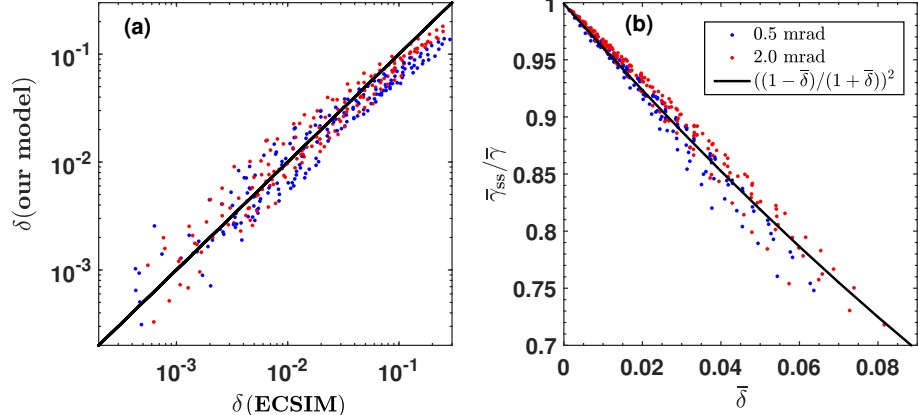

**Figure 3.** (a) Comparison of the volume linear depolarization ratios $\delta(z)$ computed with our analytical MS model and computed with ESA's Monte-Carlo model ECSIM (more details in the text, 1:1 line is given as solid diagonal line) and (b) comparison of our computations of the relationship between the single-scattering-to-total-scattering-attenuated-backscatter ratio $\overline{\gamma}_{ss}/\overline{\gamma}$ and the cloud-integrated depolarization ratio $\overline{\delta}$ (red and black circles) with the respective values for this relationship as retrieved from CALIPSO multiple scattering observations (solid black line). For the two different FOVs (0.5 mrad in black, 2.0 mrad in red) $4\times4$ $R_e(z_{ref})$ - $\alpha(z_{ref})$ combinations are considered together with different cloud penetration depths $\Delta z_{ref}$ from 10 to 70 m (with 10 m step width). All in all more than 200 simulations are included in each of the panels (a) and (b).





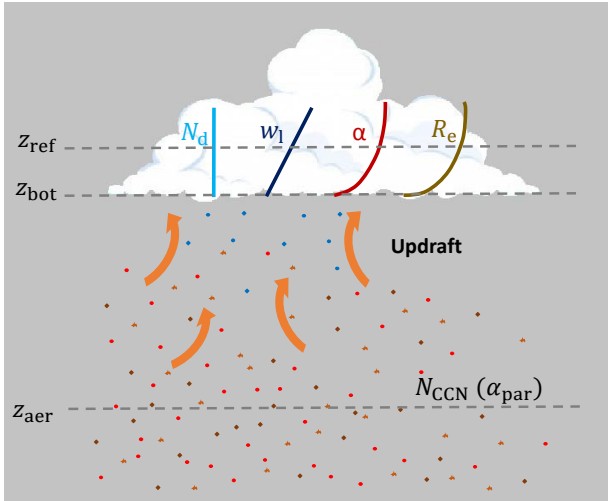

**Figure 4.** Illustration of the overall concept to investigate aerosol-cloud interaction by combining observations of cloud microphysical properties at height $z_{\mathrm{ref}}$ 50-100 m above cloud base at $z_{\mathrm{bot}}$ with aerosol properties (particle extinction coefficient $\alpha_{\mathrm{par}}$, cloud condensation nucleus concentration $N_{\mathrm{CCN}}$) measured at height $z_{\mathrm{aer}}$ several 100 m below cloud base. The indicated height profiles of cloud microphysical properties are used in the simulations to develop the new cloud retrieval scheme. Subadiabatic conditions in the lowest part of the cloud layer are assumed with an height-independent droplet number concentration $N_{\mathrm{d}}(z)$ and a linearly increasing liquid-water content $w_{\mathrm{l}}(z)$. The profiles of the cloud extinction coefficient $\alpha(z)$ and the droplet effective radius $R_{\mathrm{e}}(z)$ are then computed with Eqs. (23) and (19), respectively. All cloud parameters are zero at cloud base.

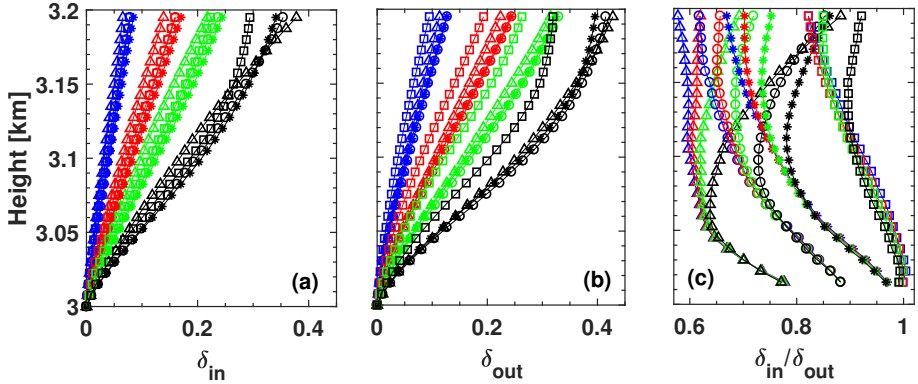

**Figure 5.** Simulated depolarization ratio profiles for (a) $\mathrm{FOV}_{\mathrm{in}}$ of 1 mrad, (b) $\mathrm{FOV}_{\mathrm{out}}$ of 2 mrad, and (c) profiles of the ratio $\delta_{\mathrm{in}}(z)/\delta_{\mathrm{out}}(z)$. $\alpha(z_{\mathrm{ref}})$ values are 5.2 km$^{-1}$ (blue), 10.4 km$^{-1}$ (red), 15.6 km$^{-1}$ (green), and 26.0 km$^{-1}$ (black) (see Table 1, $z_{\mathrm{ref}}$ =75 m above cloud base at $z_{\mathrm{bot}}$ = 3 km). Different symbols indicate different simulated $R_{\mathrm{e}}(z_{\mathrm{ref}})$ values (3.6 $\mu$m (triangle), 5.8 $\mu$m (circle), 7.9 $\mu$m (star), and 14.4 $\mu$m (square)). A clear dependence of $R_{\mathrm{e}}(z_{\mathrm{ref}})$ on $\delta_{\mathrm{in}}(z)/\delta_{\mathrm{out}}(z)$ is visible up to about 100 m above cloud base.




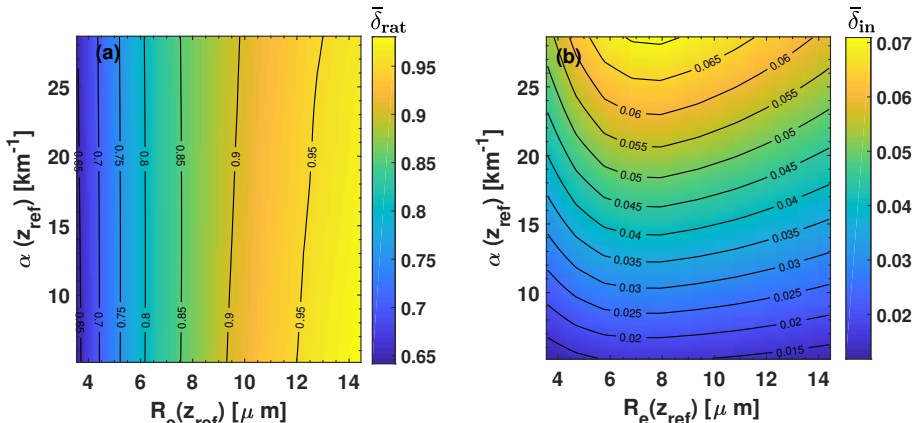

**Figure 6.** (a) Ratio $\overline{\delta}_{\mathrm{rat}}(z_{\mathrm{bot}}, z_{\mathrm{ref}})$ of depolarization ratios (see Eq. (27), integration height range of $\Delta z_{\mathrm{ref}}$=75 m) as a function of droplet effective radius $R_{\mathrm{e}}(z_{\mathrm{ref}})$ and cloud extinction coefficient $\alpha(z_{\mathrm{ref}})$. Cloud base $z_{\mathrm{bot}}$ is at 3 km height, $z_{\mathrm{ref}}$ is thus at 3.075 km height. (b) Integrated depolarization ratio $\overline{\delta}_{\mathrm{in}}(z_{\mathrm{bot}}, z_{\mathrm{ref}})$ as a function of $R_{\mathrm{e}}(z_{\mathrm{ref}})$ and $\alpha(z_{\mathrm{ref}})$. Isolines of $\overline{\delta}_{\mathrm{rat}}$ in (a) show the strong dependence of $\overline{\delta}_{\mathrm{rat}}$ on the effective radius. The $\overline{\delta}_{\mathrm{in}}$ isolines in (b) highlight the dominating influence of the extinction coefficient on $\overline{\delta}_{\mathrm{in}}$. The figures are based on 720 simulated cloud scenarios for each of the FOVs of 1 mrad and 2 mrad.

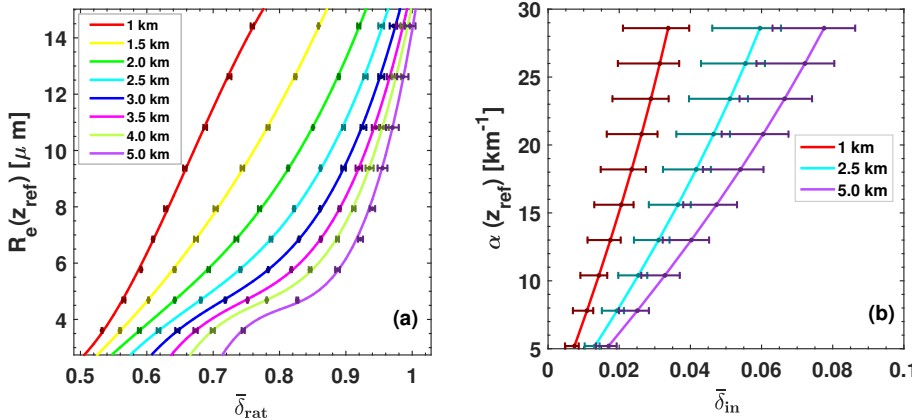

**Figure 7.** (a) Droplet effective radius $R_{\mathrm{e}}(z_{\mathrm{ref}})$ as a function of $\overline{\delta}_{\mathrm{rat}} = \overline{\delta}_{\mathrm{in}}/\overline{\delta}_{\mathrm{out}}$ for $\Delta z_{\mathrm{ref}}$=75 m and (b) relationship between the measured $\overline{\delta}_{\mathrm{in}}$ for FOV=1 mrad and cloud extinction coefficient $\alpha$. Eight different cloud layers with base height $z_{\mathrm{bot}}$ from 1–5 km height (given as numbers in the panels) are simulated in (a), three different layers are simulated in (b). For each cloud layer (indicated by different colors) simulations with all combinations of $R_{\mathrm{e}}$-$\alpha$ profile pairs (in Table 1) are performed. The small bars in (a) indicate the range of possible $\delta_{\mathrm{rat}}$ values for a given $R_{\mathrm{e}}$ value and the length of the bars indicate the very low $\alpha$ influence on the $R_{\mathrm{e}}$ retrieval (simulated $\alpha$ range is given in Table 1). A polynomial regression is applied to the mean values of $\overline{\delta}_{\mathrm{rat}}$. This regression analysis is performed for each of the eight cloud layers. The cubic model (Eq. 28) for each cloud layer is indicated as thick solid colored line. The bars in (b) indicate the range of possible $\delta_{\mathrm{in}}$ values for a given $\alpha$ value. Here, the length of the bars indicate the relatively strong $R_{\mathrm{e}}$ influence on the $\alpha(z_{\mathrm{ref}})$ retrieval (simulated $R_{\mathrm{e}}$ range is given in Table 1). The respective regression analysis leads here to the thick solid lines calculated with Eq. (29).

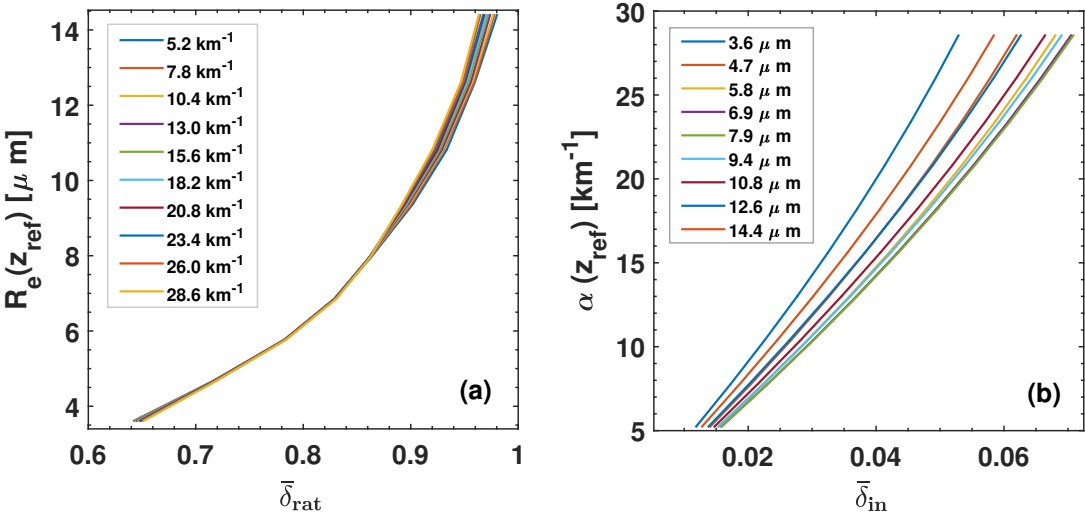

**Figure 8.** Two-step approach to derive $R_e(z_{ref})$ and $\alpha(z_{ref})$ from $\overline{\delta}_{rat}(z_{bot}, z_{ref})$ and $\overline{\delta}_{in}(z_{bot}, z_{ref})$ for a liquid cloud layer with $z_{bot}$=3 km and $z_{ref}$=3.075 km. In the first step (a), $\overline{\delta}_{rat}$ is used to determine $R_e(z_{ref})$ by means of Eq. (28), and in the second step (b), $\overline{\delta}_{in}$ and $R_e$ (from step 1) are used to determine $\alpha(z_{ref})$ with Eq. (29). In (a), all simulations with all available combinations of $R_e$-$\alpha$ profile pairs are shown to indicate the low impact of $\alpha$ (given as numbers) on the retrieval. In (b), the relationship between $\overline{\delta}_{in}$ and $\alpha$ for nine $R_e$ values (given as numbers) are shown to indicate the comparably large influence of $R_e$ on the $\alpha$ retrieval.