# Peer review of "The dual-field-of-view polarization lidar technique: A new concept in monitoring aerosol effects in liquid-water clouds — Theoretical framework"

_Atmospheric Chemistry and Physics, 2020_

## Referee Comment (RC1) · Anonymous Referee #2 · 3 Aug 2020

The paper presents and discusses a polarization-lidar technique to determine cloud microphysical parameters under subadiabatic conditions from dual field-of-view depolarization measurements. The method builds on a long tradition of multiple field-of-view lidar techniques for cloud studies that are cited in the bibliography. The authors claim that the method proposed in this paper combines both a relative simplicity and the use of strong signals (volume linear depolarization ratios), the latter allowing high time resolution in day and night conditions, which in turn makes it possible to track cloud-aerosol interaction dynamics.

The proposed technique relies on a simplified multiple scattering model that is tested by comparing its results against a more complex model and to observations supported by a sophisticated model. With the proposed technique, the droplet effective radius is first determined at a reference distance from the bottom of the cloud using the ratio of depolarization ratios at an outer (wide) and an inner (narrow) receiver field of view. Then the cloud extinction coefficient is derived from the droplet effective radius found in the previous step and the measured volume depolarization ratio at the inner field of view. Subadiabatic condition in the lowest part of the cloud are assumed to retrieve droplet number concentration and the liquid water content profile from the cloud base.

The paper – the first of a series of two, the second one presenting the results of case studies using the technique discussed in this one – is well and clearly written, and it supports convincingly the originality and the potential practicability of the method, which is supposed to be demonstrated with the case studies to be presented in the second paper of the series. I would only suggest some minor additions and modifications:

1. The final pair of field of views proposed by the authors seem to be 1 mrad and 2 mrad. The authors also state that, after performing simulations, with different fields of view (FOV), "the highest sensitivity (optimum pair of FOVs) is given for the case with the highest $FOV_{out}$-to-$FOV_{in}$ ratio". While it can be understood that the maximum FOV ($FOV_{out}$) may be limited for the sake of exploring a horizontally homogenous zone of the cloud, one would expect more explanations on the reason why $FOV_{in}$ is chosen as 1 mrad, and not a smaller value. The term "optimum" in the quoted sentence (line 28, page 11) is perhaps not accurate in this case. In my understanding and optimum pair would mean that every other pair shows worse performance, which I'm not sure is what the authors mean.
2. The authors also state that their "data analysis scheme [] will deliver the cloud microphysical products for height $z_{ref}$ that is 50–100 m above cloud base height $z_{bot}$" (lines 28-29, page 9). However in the later analysis they choose $z_{ref} = 75$ m. Some explanation on the considerations for this choice would also be helpful.
3. I would suggest that the explanation on the polynomial fitting to the simulations of $R_e(z_{ref})$ as a function of $\overline{\delta}_{rat}$ in the caption of fig 7 is moved to or repeated in the main text.
4. In line 14 of page 11, the authors say that the "striking feature in Fig. 5 is the clear dependence of the droplet effective radius $R_e(z_{ref})$ on $\delta_{in}/\delta_{out}$". While from the formal point of view this is correct, I think it would be more "physical" to say that it is $\delta_{in}/\delta_{out}$ what depends on $R_e(z_{ref})$.

5. I found the last two sentences of section 7 (Summary) ("The field site of Punta Arenas..., etc.") somewhat misplaced. Earlier in the paragraph, the authors already explain that the technique introduced is being used in a field campaign in Punta Arenas. If the quoted sentences are intended to highlight the interest of the campaign, I think it would be better placed right after the first mention to the campaign.

6. The function $n(r)$, the radius distribution function of droplet number concentration, used in Eq. (1) and subsequent equations, is never defined. Although its meaning is clear, I would strongly suggest to define it.

Other minor remarks follow:

1. Page 2, line 6: "the become cloud droplets" should probably be "to become cloud droplets".

2. Page 2, line 29: "was however". Do the authors mean "was therefore"?

3. In page 5, line 18, the cloud extinction coefficient and the droplet effective radius are called observables: "In the next sections, we evaluate the possibilities of retrieving information about these two observable parameters". It's perhaps a matter of definition and context, but, in the paper context, can a parameter be called an observable parameter when the retrieval of information on it from cloud depolarization measurements is under evaluation? In the paper context, the observables are, in my opinion, the depolarization ratios. Admittedly, this is a debatable issue and I don't absolutely oppose to the use of observable, in the sense of a physical quantity that can be measured, applied to the cloud extinction coefficient and the droplet effective radius; I wish just to make aware the authors of a possible overuse of the term.

4. Page 5, lines 26-27: "rotations of the polarization plane of the laser pulse will occur". It would probably be more accurate to write "rotations of the polarization plane with respect to that of the laser pulse will occur".

5. Page 6, line 23: "can not" should probably be "cannot".

6. Page 7, line 22: "After presenting the principle relationship". Do the authors mean "principle" or "principal"?

7. The paragraph after Eq. (12) defining the parameters appearing in Eqs. (10) and (11), should probably better placed right after Eq. (11).

8. Page 8, lines 18-19: "second mirror" should probably be "secondary mirror.

9. Page 9, line 19: "well describes". Probably "describes well" is a better construction.

10. Page 9, line 29: "for height $z_{ref}$ that is 50–100 m above cloud base height $z_{bot}$". I would suggest "for a height $z_{ref}$ that is 50–100 m above the cloud base height $z_{bot}$"

11. The pass from Eq. (21) to Eq. (24) is straightforward. The intermediate equations could probably be dropped without impairing the manuscript intelligibility.

12.  Page 11, line 1: "The profile of $\alpha(z)$ is shown in Fig. 4 as well". I would say that the profile of $\alpha(z)$ is sketched, rather than shown, in Fig. 4.

13.  Page 12, paragraph starting in line 10. It should be made clear that the references to Fig. 7 are more specifically to Fig. 7a.

14.  Page 13, line 11 "we conclude that" is repeated (already said in the preceding line).

15.  Page 13, last line: "POortable" should probably be "POrtable".

16.  Page 14, line 29: "The parameterization hold" should be "The parameterization holds"

---

## Referee Comment (RC2) · Anonymous Referee #1 · 4 Aug 2020

This paper presents the theoretical framework for a lidar retrieval of liquid water cloud extinction coefficient and droplet effective radius. These two quantities can then be used to derive estimates of cloud liquid water content and droplet number concentration. This offers the intriguing possibility of studying aerosol-cloud interactions at high temporal resolution using co-located retrievals of both aerosol and cloud particle number concentrations from ground-based lidar. Assumptions of the method are clearly described. A second, companion, paper applies these retrievals to lidar observations. The paper is well organized and well written but, in a few places, the text is not clear. These areas should be clarified:

[Figure]

1) In section 3.3, it is not clear what the multiple scattering model is. Is line 30 on page 7 saying that the Zege small-angle solution is being used? A few more sentences of description of the model – its approach and limitations – would be helpful.

2) Page 9, line 26: Given that not all readers may be familiar with the concept of subadiabatic cloud, it would be good to explain the term, its relevance to the retrieval problem, and how the accuracy of the retrieval will be impacted when this assumption is violated.

3) On page 15, line 3 mentions an uncertainty of 50%. Is this the uncertainty of the retrieved extinction, CCN concentration, or both?

4) The Polly instrument is mentioned several times, but never described. It was not clear if the Polly lidar is an HSRL or a standard backscatter depolarization lidar. A short description of the Polly instrument intended for this retrieval would be helpful.

The attached file contains a few edits to correct places where the English sounds a little odd.

Please also note the supplement to this comment:
https://acp.copernicus.org/preprints/acp-2020-473/acp-2020-473-RC2-supplement.pdf

**Supplement:**

[revised manuscript text omitted]

$$\pm\sigma_{ran,R_e}(\Delta\overline{\delta}_{rat}) = R_e \pm \left( R_0 + R_1 \times (\overline{\delta}_{rat} \pm \Delta\overline{\delta}_{rat}) + R_2 \times (\overline{\delta}_{rat} \pm \Delta\overline{\delta}_{rat})^2 + R_3 \times (\overline{\delta}_{rat} \pm \Delta\overline{\delta}_{rat})^3 \right) \tag{30}$$

and by taking half of the respective uncertainty bars.

Systematic retrieval uncertainties $\sigma_{sys,R_e}$ arise from the use of the model (polynomial functions in Fig. 7), from the uncertainties in the determined cloud base height $\Delta z_{bot}$ (we assume $\pm 15$ m), and the influence of the cloud extinction coefficient (the uncertainty is denoted here as $\Delta\alpha$ and given by the range of values in Table 1 from 5.2 to 28.6 Mm$^{-1}$). From the extended error simulations and from the analysis with real (observational) data we conclude that

we conclude that

$$\sigma_{sys,R_e}(\Delta\alpha) \approx 0.15 R_e(z_{ref}), \tag{31}$$

$$\sigma_{sys,R_e}(\Delta z_{bot}) \approx 0.10 R_e(z_{ref}). \tag{32}$$

On average, input uncertainties may partly cancel out and the mean uncertainty is given by

$$\sigma_{sys,R_e}(\Delta\alpha, \Delta z_{bot}) = \sqrt{\sigma_{sys,R_e}(\Delta\alpha)^2 + \sigma_{sys,R_e}(\Delta z_{bot})^2}. \tag{33}$$

The influence of measurement uncertainties on the retrieval of $\alpha(z_{ref})$ is estimated by considering the standard deviation $\pm\Delta\overline{\delta}_{in}$ in the computation,

$$\pm\sigma_{ran,\alpha}(\Delta\overline{\delta}_{rat}) = \alpha \pm \left( \alpha_0 + \alpha_1 \times (\overline{\delta}_{in} \pm \Delta\overline{\delta}_{in}) + \alpha_2 \times (\overline{\delta}_{in} \pm \Delta\overline{\delta}_{in})^2 \right). \tag{34}$$

In a similar way as described above for the systematic uncertainty in $R_e$, we estimated $\sigma_{sys,\alpha}$ with $\Delta z_{bot} \pm 15$ m and by using $\Delta R_e$ according to Eq. (33) in the second retrieval step to obtain $\alpha(z_{ref})$. Again, from many simulations we concluded that

$$\sigma_{sys,\alpha}(\Delta R_e) \approx 0.08 \alpha(z_{ref}), \tag{35}$$

$$\sigma_{sys,\alpha}(\Delta z_{bot}) \approx 0.15 \alpha(z_{ref}). \tag{36}$$

The overall mean systematic uncertainty may be given by:

$$\sigma_{sys,\alpha}(\Delta R_e, \Delta z_{bot}) = \sqrt{\sigma_{sys,\alpha}(\Delta R_e)^2 + \sigma_{sys,\alpha}(\Delta z_{
[revised manuscript text omitted]

---

## Author Comment (AC2) · 6 Oct 2020

Dear Editor, Dear Reviewers,

Many thanks for your time and efforts.

Please find our reply letter in the supplement,

best wishes,

Cristofer Jimenez (on behalf of all authors)

Please also note the supplement to this comment:

https://acp.copernicus.org/preprints/acp-2020-473/acp-2020-473-AC2-supplement.pdf

---

## Author Comment (AC1)

**Author response to RC1 and RC2 (acp-2020-473)**

**Dear Editor,**

We thank the reviewers for careful reading and constructive suggestions. In this document we would like provide our answers to both Reviews.

Our answers are marked in bold text. In the revised version of the manuscript all changes due to the expressed suggestions were highlighted using different colors. Blue for changes due to Referee comment RC1, and red for changes due to RC2.

Additionally, we would like to make two small changes in the manuscript. First, to make the explanations straighter forward, we removed Equation (7) and Figure 1.a, since they really do not provide essential information to the explanations given in Section 3.2. Additionally we updated the values inside Table 2. The previous values were generated using another definition of the integrated depolarization ratio, i.e. integrating directly the depolarization ratios instead of the signals, as it is described in section 4. This changes do not affect any result and conclusion provided in the manuscript.

Review RC1 Anonymous Referee #2 (blue)

The paper presents and discusses a polarization-lidar technique to determine cloud microphysical parameters under subadiabatic conditions from dual field-of-view depolarization measurements. The method builds on a long tradition of multiple field-of-view lidar techniques for cloud studies that are cited in the bibliography. The authors claim that the method proposed in this paper combines both a relative simplicity and the use of strong signals (volume linear depolarization ratios), the latter allowing high time resolution in day and night conditions, which in turn makes it possible to track cloud-aerosol interaction dynamics.

The proposed technique relies on a simplified multiple scattering model that is tested by comparing its results against a more complex model and to observations supported by a sophisticated model. With the proposed technique, the droplet effective radius is first determined at a reference distance from the bottom of the cloud using the ratio of depolarization ratios at an outer (wide) and an inner (narrow) receiver field of view. Then the cloud extinction coefficient is derived from the droplet effective radius step and the measured volume depolarization ratio at the

inner field of view. Subadiabatic condition in the lowest part of the cloud are assumed to retrieve droplet number concentration and the liquid water content profile from the cloud base. The paper – the first of a series of two, the second one presenting the results of case studies using the technique discussed in this one – is well and clearly written, and it supports convincingly the originality and the potential practicability of the method, which is supposed to be demonstrated with the case studies to be presented in the second paper of the series. I would only suggest some minor additions and modifications:

1. The final pair of field of views proposed by the authors seem to be 1 mrad and 2 mrad. The authors also state that, after performing simulations, with different fields of view (FOV), "the highest sensitivity (optimum pair of FOVs) is given for the case with the highest FOVout-to-FOVin ratio". While it can be understood that the maximum FOV (FOVout) may be limited for the sake of exploring a horizontally homogenous zone of the cloud, one would expect more explanations on the reason why FOVin is chosen as 1 mrad, and not a smaller value. The term "optimum" in the quoted sentence (line 28, page 11) is perhaps not accurate in this case. In my understanding and optimum pair would mean that every other pair shows worse performance, which I'm not sure is what the authors mean.

Yes, we used the term optimum only considering the sensitivity, but right as you said, because of the effect of cloud inhomogeneities, the pair of FOVs with the largest contrast would not be the optimal choice. This can be seen in the Figure below, which sketch the sensitivity for different pair of FOVs at different cloud base height (left) and the mean sensitivity vs the ratio between FOV-in and FOV-out. This figure was excluded from the manuscript to keep it simpler.

We used in our approach this FOV-divergence values (1 and 2 mrad) because most of standard polarization lidars, such as the Polly systems (Engelmann et al. 2016) and Earlinet systems (Pappalardo et al. 2014) already possess a 1.0 mrad receiver-FOV. Most of those systems possess additionally a near range total scattering receiver at a second FOV, which allow an instrumental upgrade only by adding just one cross-polarized detection channel.

We removed the word 'optimum' from the text and added a short explanation about the choice of FOVs. To check whether the horizontal inhomogeneities affect our approach in one real case, we considered one of the measurement examples presented in part 2.

The authors also state that their "data analysis scheme will deliver the cloud microphysical products for height zref that is 50–100 m above cloud base height zbot" (lines 28-29, page 9). However in the later analysis they choose zref = 75 m. Some explanation on the considerations for this choice would also be helpful.

In the paper we present the case when we choose 75 meters penetration depth. From Figure 5 one can see the dependence of the ratio of depolarization values ( $\delta_{in}/\delta_{out}$ ) depends exclusively on the size (and not on the extinction) in the first part of the cloud, being this dependency lost after the first 100 meters. In principle one can apply the approach for each height bin between 50 and 100 meters, but for the sake of simplicity, in the manuscript we limited the scheme to the case of 75m. We state that now on page 10, after Eq.(14).

3. I would suggest that the explanation on the polynomial fitting to the simulations of  $R_e(z_{ref})$  as a function of  $\delta_{rat}$  in the caption of fig 7 is moved to or repeated in the main text.

**Done**

4. In line 14 of page 11, the authors say that the "striking feature in Fig. 5 is the clear dependence of the droplet effective radius Re(zref) on  $\delta in/\delta out$ ". While from the formal point of view this is correct, I think it would be more "physical" to say that it is  $\delta in/\delta out$  what depends on Re(zref).

You are right, although for the inversion perspective is the effective radius threated as depending on the measured ratio  $\delta_{in}/\delta_{out}$ . Physically speaking, it is the ratio which depends on the effective radius. We rearranged the sentence.

5. I found the last two sentences of section 7 (Summary) ("The field site of Punta Arenas..., etc.") somewhat misplaced. Earlier in the paragraph, the authors already explain that the technique introduced is being used in a field campaign in Punta Arenas. If the quoted sentences are intended to highlight the interest of the campaign, I think it would be better placed right after the first mention to the campaign.

After looking again at the paragraph we also find that the description of the situation at the field site is better placed right after mentioning the campaign. We rearrange the paragraph.

6. The function n(r) the radius distribution function of droplet number concentration, used in Eq. (1) and subsequent equations, is never defined. Although its meaning is clear, I would strongly suggest to define it.

Yes, since the parameter is well known, we don't see a need to define it in detail. We would just add the notation in the text after mentioning it.

i.e.:

$$w_{\rm l} = \frac{4}{3}\pi\rho_{\rm w} \int_{0}^{\infty} n(r)r^3 dr = \frac{4}{3}\pi\rho_{\rm w} \left(\frac{\int_{0}^{\infty} n(r)r^3 dr}{\int_{0}^{\infty} n(r)dr}\right) \int_{0}^{\infty} n(r)dr = \frac{4}{3}\pi\rho_{\rm w} R_{\rm v}^3 N_{\rm d}$$
(1)

with the total droplet number concentration  $N_d = \int_0^\infty n(r)dr$ , the volume mean droplet radius  $R_v$  of a given droplet size distribution n(r), and the liquid-water density  $\rho_w$ .

**and then to continue with the sentence in blue: The droplet number concentration n(r) is described by a modified gamma size distribution (see Eq.(2) in Schmidt et al., 2014).**

Other minor remarks follow:

1. Page 2, line 6: "the become cloud droplets" should probably be "to become cloud droplets". **Done**

2. Page 2, line 29: "was however". Do the authors mean "was therefore"? Done

3. In page 5, line 18, the cloud extinction coefficient and the droplet effective radius are called observables: "In the next sections, we evaluate the possibilities of retrieving information about these two observable parameters". It's perhaps a matter of definition and context, but, in the paper context, can a parameter be called an observable parameter when the retrieval of information on it from cloud depolarization measurements is under evaluation? In the paper context, the observables are, in my opinion, the depolarization ratios. Admittedly, this is a debatable issue and I don't absolutely oppose to the use of observable, in the sense of a physical quantity that can be measured, applied to the cloud extinction coefficient and the droplet effective radius; I wish just to make aware the authors of a possible overuse of the term.

Yes, thanks! It may depends on who is dealing with the approach. In absolute sense, the depolarization ratios are the observables. Although for analysis based on our retrievals the cloud properties may be treated as the observables... To keep it consistent, in the paper, we will use the term observable only for the depolarization ratios, and the clouds parameters as what they are, a derived (or retrieved) product, so we modified the text in page 5, line 18.

4. Page 5, lines 26-27: "rotations of the polarization plane of the laser pulse will occur". It would probably be more accurate to write "rotations of the polarization plane with respect to that of the laser pulse will occur". **Indeed, yes.**

5. Page 6, line 23: "can not" should probably be "cannot". Done

6. Page 7, line 22: "After presenting the principle relationship". Do the authors mean "principle" or "principal"?

Here, with "principle", we intended to refer to the basic relationship between depolarization and cloud-microphysics based on our simple model that just consider the single processes of forward and backward scattering.

7. The paragraph after Eq. (12) defining the parameters appearing in Eqs. (10) and (11), should probably better placed right after Eq. (11). **Done**

8. Page 8, lines 18-19: "second mirror" should probably be "secondary mirror. Done

9. Page 9, line 19: "well describes". Probably "describes well" is a better construction. Done

10. Page 9, line 29: "for height zref that is 50–100 m above cloud base height zbot". I would suggest "for a height zref that is 50–100 m above the cloud base height zbot" **Done**

11. The pass from Eq. (21) to Eq. (24) is straightforward. The intermediate equations could probably be dropped without impairing the manuscript intelligibility.

**Yes, it is straight forward the derivation. We drop Eq. (22) but leave Eq. (23) to make clear the usage of a reference value to define Eq. (24). (now Eq. (23))**

12. Page 11, line 1: "The profile of  $\alpha(z)$  is shown in Fig. 4 as well". I would say that the profile of  $\alpha(z)$  is sketched, rather than shown, in Fig. 4. **Done**

13. Page 12, paragraph starting in line 10. It should be made clear that the references to Fig. 7 are more specifically to Fig. 7a. **Done**

- 14. Page 13, line 11 "we conclude that" is repeated (already said in the preceding line). Done
- 15. Page 13, last line: "POortable" should probably be "POrtable". Done

16. Page 14, line 29: "The parameterization hold" should be "The parameterization holds" Done

All corrections and suggestions were carefully adopted in the manuscript. We are grateful for them.

**References:**

Engelmann, R., Kanitz, T., Baars, H., Heese, B., Althausen, D., Skupin, A., Wandinger, U., Komppula, M., Stachlewska, I. S., Amiridis, V., Marinou, E., Mattis, I., Linné, H., and Ansmann, A.: The automated multiwavelength Raman polarization and water-vapor lidar PollyXT: the neXT generation, Atmos. Meas. Tech., 9, 1767–1784, https://doi.org/10.5194/amt-9-1767-2016, 2016.

Pappalardo, G., Amodeo, A., Apituley, A., Comeron, A., Freudenthaler, V., Linné, H., Ansmann, A., Bösenberg, J., D'Amico, G., Mattis, I., Mona, L., Wandinger, U., Amiridis, V., Alados-Arboledas, L., Nicolae, D., and Wiegner, M.: EARLINET: towards an advanced sustainable European aerosol lidar network, Atmos. Meas. Tech., 7, 2389–2409, doi:10.5194/amt-7-2389-2014, 2014.

**Review RC2 Anonymous Referee #1 (RED)**

This paper presents the theoretical framework for a lidar retrieval of liquid water cloud extinction coefficient and droplet effective radius. These two quantities can then be used to derive estimates of cloud liquid water content and droplet number concentration. This offers the intriguing possibility of studying aerosol-cloud interactions at high temporal resolution using co-located retrievals of both aerosol and cloud particle number concentrations from ground-based lidar. Assumptions of the method are clearly described. A second, companion, paper applies these retrievals to lidar observations. The paper is well organized and well written but, in a few places, the text is not clear. These areas should be clarified:

1) In section 3.3, it is not clear what the multiple scattering model is. Is line 30 on page 7 saying that the Zege small-angle solution is being used? A few more sentences of description of the model – its approach and limitations – would be helpful.

As you concluded: the Zege small-angle solution is used in our work.

In same page line 24 it says: After presenting the principle relationship between the measured linear depolarization ratio, forward scattering, and droplet size, next we introduce the multiple scattering model used to develop our retrieval method presented in Sect. 4. And then in line 32: The so-called small-angle approximation is adopted in this work. But this does not clarify whether it is used in ours or Zege's work.

The information is there but as you say, it is not clear enough. We rephrased the paragraph to avoid confusions. We also included additional information about the approach and its limitations. Please see how the changes in response to this question and the next ones are highlighted in red in the revised version of the manuscript.

2) Page 9, line 26: Given that not all readers may be familiar with the concept of subadiabatic cloud, it would be good to explain the term, its relevance to the retrieval problem, and how the accuracy of the retrieval will be impacted when this assumption is violated.

The sub-adiabatic cloud model simply considers a reduced water content due to entrainment of dry air masses from above the cloud. According to Merk et al 2016, the water content profiles under sub-adiabatic conditions can be expressed as follows:

 $w_l = \Gamma \Delta z = f_{ad} \Gamma_{ad}(p, T) \Delta z$

Here  $f_{ad}$  refers to the so-called adiabatic factor (or degree of adiabaticity) and it can be larger than zero and smaller than one and  $\Gamma_{ad}(p,T)$  refers to the adiabatic lapse rate. Whether the system is adiabatic or sub-adiabatic does not affect the fact that the water content increase linearly.

Thanks for this suggestion. The term sub-adiabatic has not been much widely used so we added a short explanation about the term. We also provide some sentences on the impact of this in our retrieval in section 5.

3) On page 15, line 3 mentions an uncertainty of 50%. Is this the uncertainty of the retrieved extinction, CCN concentration, or both?

The uncertainties of 50% are on the CCN concentrations. This has been revealed by different comparison studies from the CCN concentration estimates between ground base remote sensing and in-situ aircraft measurements (Düsing et al., 2018; Haarig et al., 2019). We modified the text to make this clearer.

4) The Polly instrument is mentioned several times, but never described. It was not clear if the Polly lidar is an HSRL or a standard backscatter depolarization lidar. A short description of the Polly instrument intended for this retrieval would be helpful. The attached file contains a few edits to correct places where the English sounds a little odd.

A more detailed description of the instrument was originally provided only in part II. We see however the value of a short description of the system in part I, so we added some information about it, e.g. ... *the Raman polarization lidar Polly*....

We appreciate the suggested corrections about the language usage. We have employed them all in the revised version of the manuscript. Thanks a lot.

**References:**

Düsing, S., Wehner, B., Seifert, P., Ansmann, A., Baars, H., Ditas, F., Henning, S., Ma, N., Poulain, L., Siebert, H., Wiedensohler, A., and Macke, A.: Helicopter-borne observations of the continental background aerosol in combination with remote sensing and ground-based measurements, Atmos. Chem. Phys., 18, 1263–1290, https://doi.org/10.5194/acp-18-1263-2018, 2018.

Haarig, M., Walser, A., Ansmann, A., Dollner, M., Althausen, D., Sauer, D., Farrell, D., and Weinzierl, B.: Profiles of cloud condensation nuclei, dust mass concentration, and ice-nucleating-particlerelevant aerosol properties in the Saharan Air Layer over Barbados from polarization lidar and airborne in situ measurements, Atmos. Chem. Phys., 19, 13773–13788, https://doi.org/10.5194/acp-19-13773-2019, 2019.

Merk, D., Deneke, H., Pospichal, B., and Seifert, P.: Investigation of the adiabatic assumption for estimating cloud micro- and macrophysical properties from satellite and ground observations, Atmos. Chem. Phys., 16, 933–952, https://doi.org/10.5194/acp-16-933-2016, 2016.